EMBO
Molecular Medicine

# Safe engineering of CAR T cells for adoptive cell therapy of cancer using long-term episomal gene transfer

Chuan Jin, Grammatiki Fotaki[‡], Mohanraj Ramachandran[‡], Berith Nilsson, Magnus Essand[\*,†] & Di Yu[†,\*\*]

## Abstract

Chimeric antigen receptor (CAR) T-cell therapy is a new successful treatment for refractory B-cell leukemia. Successful therapeutic outcome depends on long-term expression of CAR transgene in T cells, which is achieved by delivering transgene using integrating gamma retrovirus (RV) or lentivirus (LV). However, uncontrolled RV/LV integration in host cell genomes has the potential risk of causing insertional mutagenesis. Herein, we describe a novel episomal long-term cell engineering method using non-integrating lentiviral (NILV) vector containing a scaffold/matrix attachment region (S/MAR) element, for either expression of transgenes or silencing of target genes. The insertional events of this vector into the genome of host cells are below detection level. CD19 CAR T cells engineered with a NILV-S/MAR vector have similar levels of CAR expression as T cells engineered with an integrating LV vector, even after numerous rounds of cell division. NILV-S/MAR-engineered CD19 CAR T cells exhibited similar cytotoxic capacity upon CD19[+] target cell recognition as LV-engineered T cells and are as effective in controlling tumor growth *in vivo*. We propose that NILV-S/MAR vectors are superior to current options as they enable long-term transgene expression without the risk of insertional mutagenesis and genotoxicity.

**Keywords** CAR T cells; episomal cell engineering; non-integrating lentivirus (NILV); scaffold/matrix attachment region (S/MAR) element; self-replicating DNA
**Subject Categories** Cancer; Genetics, Gene Therapy & Genetic Disease; Immunology

## Introduction

Cancer immunotherapy using T cells engineered *ex vivo* with a chimeric antigen receptor (CAR) against CD19 is currently facing major breakthroughs in the treatment of B-cell malignancies (Davila *et al*, 2014; Maude *et al*, 2014; Kochenderfer *et al*, 2015; Lee *et al*, 2015). Integrating gamma retroviral (RV) or lentiviral (LV) vectors are then used for CD19 CAR engineering of T cells. The safety level of these vectors is today high, achieved by partial deletion of the U3 region of 3′ long terminal repeat (LTR) and the use of, for example, a cytomegalovirus (CMV) promoter to replace the U3 region at the 5′ LTR for initial transcription. This strategy drastically reduces the transcriptional activity from virus LTR (Iwakuma *et al*, 1999). However, the uncontrolled integration can still cause insertional mutagenesis and lead to overexpression of adjacent genes or disruption of genes at the site of integration. This raises the concern that RV/LV-engineered T cells can become tumorigenic (Baum *et al*, 2003; Modlich *et al*, 2009). To our knowledge, it is not yet been observed. However, insertional mutagenesis-driven clonal dominance or oncogenesis has been observed in RV-engineered hematopoietic stem cells in clinical gene therapy trials for X-linked severe combined immune deficiency (Hacein-Bey-Abina *et al*, 2008), chronic granulomatous disease (Ott *et al*, 2006), and Wiskott–Aldrich syndrome (Avedillo Diez *et al*, 2011).

Non-integrating lentiviral vectors (NILVs) have been developed to reduce insertional mutagenesis-induced genotoxicity by eliminating the integrase function of LV vectors (Apolonia *et al*, 2007). Cells transduced with a NILV have double-stranded DNA circles accumulated inside the nucleus facilitating transgene expression (Farnet & Haseltine, 1991; Butler *et al*, 2001). However, this type of episomal DNA is diluted upon cell division, which means that CAR T cells engineered with NILV will eventually lose their CAR expression.

Researchers have over the past decades developed a series of non-viral plasmids (pEPI series) containing a scaffold/matrix attachment region (S/MAR) element for episomal long-term expression of

Department of Immunology, Genetics and Pathology, Science for Life Laboratory, Uppsala University, Uppsala, Sweden
  *Corresponding author. Tel: +46 18 471 4535; E-mail: magnus.essand@igp.uu.se
  **Corresponding author. Tel: +46 18 471 4524; E-mail: di.yu@igp.uu.se
  [†]The authors contributed equally as senior author
  [‡]The authors contributed equally as second author

transgenes (Piechaczek *et al*, 1999; Wong & Harbottle, 2013). The S/MAR is a segment of genomic DNA that anchors the chromatin to the nuclear matrix proteins and mediates structural organization of chromatin within the nucleus (Bode *et al*, 1992). S/MAR binds to scaffold attachment factor protein A (SAF-A) and provides mitotic stability of the S/MAR-containing plasmids by facilitating DNA attachment to the nuclear matrix for the segregation of DNA into progeny cells (Jenke *et al*, 2002). A new gene transfer vector termed "anchoring non-integrating lentiviral vector (aniLV)" that combined advantages of a S/MAR element and NILV was recently developed (Verghese *et al*, 2014). The authors demonstrated that aniLV can establish long-term episomal maintenance and keep transgene expression in a subset of cells including primary isolated murine hematopoietic progenitors and leukocytes. We have been working with a similar system in parallel and we hypothesize that an S/MAR-containing NILV should be an optimal vector for long-term T-cell engineering with low risk of insertional mutagenesis and genotoxicity. In this study, we demonstrate the application of NILV-S/MAR-based vectors for transgene expression, target gene down-regulation (shRNA-mediated RNA interference), and the feasibility in generating CD19 CAR T cells for cancer immunotherapy.

## Results and Discussion

### NILV-S/MAR vector exerts long-term episomal maintenance

As a proof of concept, we used green fluorescent protein (GFP) as reporter gene and designed a conventional LV vector, an integrase-deficient NILV vector, and a NILV vector with an S/MAR element as well as a NILV vector with flipped orientation of the S/MAR element (S/MAR$^{rev}$) to monitor the persistency and stability of DNA delivered to mammalian cells (Fig 1A). To investigate the ability of the NILV-S/MAR vector to stably maintain GFP expression in fast-dividing human cells, A549 (lung cancer) cells were transduced with the various vectors, cultured for 2 weeks, selected by puromycin, and monitored over time for GFP expression by flow cytometry. Cells transduced with NILV lost GFP expression rapidly due to dilution of the circularized vector during cell division, and after 14 days, almost all cells had lost GFP expression (Fig 1B and C). Almost all cells expressed GFP when transduced with the integrating LV vector and a substantial fraction of cells had persistent expression of GFP for more than 4 weeks when transduced with the non-integrating NILV-S/MAR vector (Fig 1B and C). We also observed that the orientation of the S/MAR element did not affect functionality (Fig 1C), which is in accordance with a previous report (Mielke *et al*, 1990). Hence, we decided to continue with only one orientation of the S/MAR element in the NILV-S/MAR vector design in our later studies. A similar experiment was performed on 911 cells (transformed human embryonic retinoblasts) without puromycin selection. Stable GFP expression was still observed 38 days after transduction with the NILV-S/MAR vector (Appendix Fig S1A and B).

Even though the episomal status has already been validated for the aniLV vector (Verghese *et al*, 2014), we examined extra-chromosomal existence and genome integration events of our NILV-S/MAR vector, since the design of our vector is a little different from the aniLV vector. DNA extracted from transduced and puromycin-selected A549 cells was subjected to PCR amplification of joined LTR regions (circularization). An experimental outline of the assay with primer sequences is presented in Appendix Fig S2A. PCR products were detected in DNA samples obtained only from NILV-S/MAR-transduced cells, but not from NILV-transduced or LV-transduced cells (Fig 1D). This indicates that circularized DNA was maintained in NILV-S/MAR-transduced cells but not in NILV-transduced cells lacking S/MAR, where the circularized DNA is probably lost during cell division. DNA circles are formed as an intermediate step and side product during lentivirus integration (Farnet & Haseltine, 1991; Butler *et al*, 2001) and accumulate at an elevated number inside nucleus when the integrase is mutated (Apolonia *et al*, 2007). Two types of DNA circles are formed during lentivirus integration, one where non-homologous end joining generates a 2-LTR circle and another where homologous recombination generates a 1-LTR circle (Farnet & Haseltine, 1991). NILV-S/MAR vectors formed mainly the 1-LTR circle of 552 bp (Fig 1D) and the 2-LTR circle of 1,029 bp was not detected. This is in accordance with the aniLV study that has a similar design to our constructs (Verghese *et al*, 2014). The formation of the 1-LTR circle was verified by sequencing (Appendix Fig S2B). We also studied the genomic integration events by Southern blot analysis. DNA from single cell-transduced clones was hybridized with a probe against the whole viral genome (about 6.8 kb). Appearance of a band at the same size as the positive control (viral genome digested from plasmid) proved the existence of episomal DNA circles in single cell clones from NILV-S/MAR-transduced cells (Fig 1E). We only observe a smear of DNA and no distinct band in the single cell clones from LV-transduced cells suggesting that multiple integration events occurred in these single cell clones. Since there was no other band or smear presented in NILV-S/MAR-transduced clones than the 6.8 kb band, we conclude that any possible integration event caused by the NILV-S/MAR vector was below detection level. Genomic integration was also analyzed by linear amplification-mediated PCR (LAM-PCR) as described in detail (Schmidt *et al*, 2007). Both LV and NILV-S/MAR vectors had the internal control band at 225 bp, depicting the existence of viral genome. However, only LV(GFP)-transduced cells generate amplicons (several bands around size 100 bps) corresponding to genome integration, again indicating that the NILV-S/MAR vector does not cause integrations (Appendix Fig S3). These data do not completely exclude integration events of the NILV-S/MAR vector, but indicate that integration events caused by any residual activity of mutated integrase in NILV-S/MAR vector were below detection level. It should be noted that the LAM-PCR assay only looks at LTR-directed LV integration events and do not account for other types of integration that may occur from plasmids. Furthermore, a specific padlock probe assay was developed as illustrated in Appendix Fig S4 to confirm persistence of the transgene DNA molecule. The probe specifically targets the non-transcribed region of the vector to avoid unspecific signals from mRNA binding. DNA circles colocalized with nuclei were detected in both LV-transduced cells (Fig 1F) and NILV-S/MAR-transduced cells (Fig 1H) but not in NILV-transduced cells (Fig 1G). Each red dot in Fig 1F–H represents one DNA molecule either integrated or circularized. The data also confirm that the NILV-S/MAR vector was maintained at low copy numbers (Fig 1H), as previously described for S/MAR-containing plasmids (Mielke *et al*, 1990; Jenke *et al*, 2002, 2004b).

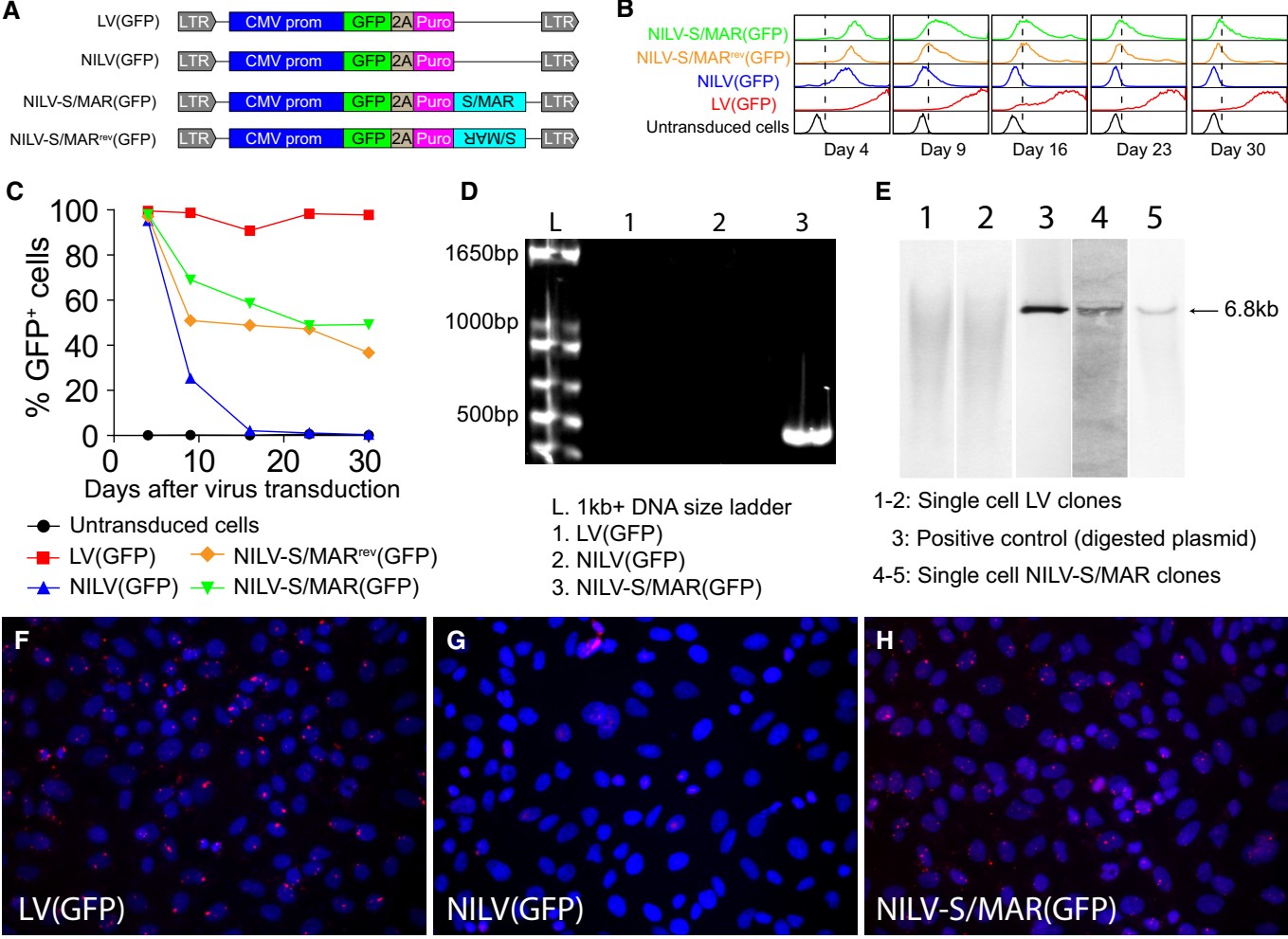

**Figure 1.    NILV-S/MAR-mediated transgene expression.**

A    Schematic drawing of the lentiviral vectors used. LV indicates conventional integrating lentiviral vector and NILV indicates non-integrating lentiviral vector, generated by using packaging plasmid with deficient integrase. S/MAR indicates that the vector contains a scaffold/matrix attachment region element and S/MAR[rev] means that the element is positioned in the reverse orientation. LTR: long terminal repeat; CMV prom: cytomegalovirus immediate early gene promoter; GFP: green fluorescent protein; 2A: viral self-cleaving T2A sequence; puro: puromycin.

B    Histograms of GFP expression (x-axis) are presented from A549 cells transduced with the different viral vectors.

C    Percentage of GFP-positive cells from the experiment described in (B).

D    A PCR assay along with primers, as illustrated in Appendix Fig S2A, was used to detect formation of LTR junction and thereby circularization of the vector. The expected size of PCR product is 552 bp from a 1-LTR circle and 1,029 bp from a 2-LTR circle.

E    Southern blot analysis to detect integration events in LV- and NILV-S/MAR-transduced single cell clones with DNA probed against the viral genome. A 6.8-kb positive control from a linearized plasmid (lane 3) was included for indication of the size of DNA circles formed by non-integrated NILV-S/MAR vector.

F–H    A padlock probe assay, as illustrated in Appendix Fig S4, was used to demonstrate persistence of proviral DNA either in a circular or in an integrated form. Representative images show DNA persistence in (F) LV-, (G) NILV-, or (H) NILV-S/MAR-transduced cells. Each red dot (Cy3) represents a single DNA molecule and cell nuclei were stained with Hoechst 33342.

Source data are available online for this figure.

## NILV-S/MAR vector for gene knockdown

We next demonstrated the possibility of using NILV-S/MAR-based vectors to achieve stable knockdown of gene expression. First, an integrating LV vector, LV(RFP-Luc2) (Fig 2A), was produced and used to create a stable A549 cell line called A549/RFP[+]Luc[+], expressing both red fluorescent protein (RFP) and codon-optimized firefly luciferase (Luc) as outlined in Fig 2B. NILV-S/MAR-based knockdown vectors with short hairpin RNA (shRNA) sequences controlled by an H1 promoter were also generated to encode an shRNA sequence targeting RFP expression, NILV-S/MAR(GFP, shRFP), and a nontarget (NT) shRNA control sequence, NILV-S/MAR(GFP, shNT) (Fig 2A). A549/RFP[+]Luc[+] cells were then transduced with the S/MAR-based knockdown vectors as outlined in Fig 2B. RFP expression was measured by flow cytometry after 20 and 27 days (Fig 2C) and luciferase expression was measured after 20 and 44 days (Fig 2D). Both assays showed stable and long-term knockdown of target genes when the NILV-S/MAR(GFP, shRFP)

vector was used. Representative fluorescence images of RFP expression are shown in Fig 2E–H and Appendix Fig S5. These findings indicate that the NILV-S/MAR-based vector system can be used to achieve stable host cell gene silencing without using an integrating viral vector or other integrating sequences. A lentiviral vector carrying an shRNA cassette is today the most efficient tool to achieve stable gene silencing in cells and this approach is frequently used to study the effects of gene silencing. One drawback with this approach, however, is that it is difficult to distinguish whether the effects on gene expression alteration are caused by knockdown of the target gene and following events or an alteration of host cell gene expression due to random integration of the shRNA cassette in the host cell genome. We therefore envision that the NILV-S/MAR-based vector system will become a useful and easy tool for safe and stable cell engineering for basic research studies.

## NILV-S/MAR vector for long-term episomal T-cell engineering

Cancer immunotherapy with genetically engineered T cells has gained increasing interests in both preclinical research and clinical applications. Many CAR T cells have been developed for adoptive therapy to cancer patients, including anti-CEA CAR T cells for colon cancer (Kershaw *et al*, 2006), anti-GD2 CAR T cells for neuroblastoma (Louis *et al*, 2011), and anti-CD19 CAR T cells for B-cell leukemia (Grupp *et al*, 2013). Here, we chose to engineer T cells with a NILV-S/MAR-based vector encoding a 2[nd] generation CAR against the CD19 molecule, NILV-S/MAR(CD19CAR), to illustrate the advantages of the technology (Fig 3A). The intracellular signaling domains of the CD19 CAR include the 4-1BB and CD3 zeta chain signaling domains (Milone *et al*, 2009). For comparison, we produced LV(CD19CAR), NILV(CD19CAR), and NILV-S/MAR(Mock) vectors (Fig 3A). The expression of CD19 CAR was monitored by flow cytometry before and after *ex vivo* expansion of T cells using the AEP protocol (Jin *et al*, 2014). T cells undergo rapid proliferation during *ex vivo* expansion. Therefore, it is an ideal platform to investigate whether the S/MAR element can maintain persistent long-term transgene expression. A schematic timeline of the experiment is illustrated in Fig 3B. T cells transduced with NILV(CD19CAR), LV(CD19CAR), or NILV-S/MAR(CD19CAR) all had CD19 CAR expression 7 days after virus transduction (Fig 3C, Before). After 12 days of T-cell expansion, we found that LV(CD19CAR)- and NILV-S/MAR(CD19CAR)-transduced T cells retained similar levels of CAR expression as before expansion (Fig 3C, After), while NILV(CD19CAR)-transduced T cells lost the CAR expression (Fig 3C, After). Representative histograms of CD19 CAR expression from one donor before and after T-cell expansion are shown in Fig 3D. We did not observe any phenotypic differences between the expanded LV- and NILV-S/MAR-engineered CD19 CAR T cells in terms of CD4:CD8 ratio (Appendix Fig S6, upper panel) or T-cell exhaustion as determined as PD1 and TIM3 double positivity (Appendix Fig S6, lower panel).

To meet clinical requirements, NILV-S/MAR-engineered CAR T cells need to be functionally similar to LV-engineered CAR T cells. To evaluate this, CD19 CAR T cells were cocultured with CD19[+] target cells (Daudi; human B lymphoblast cell line) and T-cell activity upon target cell recognition was analyzed based on CD107a expression (degranulation), secretion of IFN-γ (cytokine release), cell killing (cytotoxicity), and proliferation ability. T cells

engineered with LV(CD19CAR) and NILV-S/MAR(CD19CAR) expressed similar levels of CD107a (Fig 3E) and secreted similar amounts of IFN-γ (Fig 3F). The CD107a expression and IFN-γ secretion were significantly higher than for NILV(CD19CAR)-engineered T cells (Fig 3E and F). Furthermore, T cells engineered with LV(CD19CAR) and NILV-S/MAR(CD19CAR) displayed cytotoxic ability by killing CD19[+] target cells in a dose-dependent manner, while NILV(CD19CAR) and NILV-S/MAR(Mock) T cells did not kill target cells (Fig 3G). Both LV(CD19CAR)- and NILV-S/MAR(CD19CAR)-engineered T cells labeled with a fluorescent dye proliferated upon cognate antigen stimulation (seen as dilution of the fluorescent dye with a shift in histogram toward left) while NILV(CD19CAR)- and NILV-S/MAR(Mock)-engineered T cells did not proliferate (Fig 3H). This further confirms that NILV-S/MAR(CD19CAR)-engineered T cells possess functional and long-term CD19 CAR expression after expansion. The efficacy of NILV-S/MAR(CD19CAR)-engineered T cells to control tumor growth was examined in a xenograft mouse model with Karpas 422 tumors (human CD19[+] B-cell non-Hodgkin's lymphoma cell line). Mice treated with NILV-S/MAR(CD19CAR)-engineered T cells (Fig 3K) or LV(CD19CAR)-engineered T cells (Fig 3J) exhibited similarly suppressed tumor growth, significantly better than mice treated with NILV-S/MAR(Mock)-engineered T cells (Fig 3I and L). The survival of tumor-bearing mice was significantly prolonged when treated with CD19 CAR T cells compared to mock T cells (Fig 3M). T-cell infiltration was detected at the same levels in tumors from the NILV-S/MAR(CD19CAR) and LV(CD19CAR) treatment groups (Fig 3N–P), at significantly higher levels than in the mock-treated group (Fig 3Q). Taken together, the data suggest that the therapeutic efficacy of NILV-S/MAR-engineered CD19 CAR T cells is similar to that of T cells engineered with a self-inactivating integrating LV vector.

Apart from using RV or LV vectors for T-cell engineering, electroporation-mediated mRNA transfections are being carried out (Barrett *et al*, 2013). This approach is safer for T-cell engineering than integrating viruses, especially when the specificity of a T-cell receptor (TCR) or CAR is not fully known. Therefore, mRNA transfection can be used for investigation of toxicity and specificity of engineered T cells in early-phase clinical trials. However, mRNA electroporation does not result in a long-term transgene expression in T cells, meaning that T-cell transfer probably needs to be repeated to obtain therapeutic effects. An adenoviral hybrid vector system (Voigtlander *et al*, 2013) was recently established for nuclear delivery and episomal maintenance of the transgene. However, the gene delivery efficiency of adenoviral vector remains low in human T cells (Yu *et al*, 2013). Recently, CRISPR/Cas9 platforms have been established for site-specific genome editing, which may lead to reduced genotoxicity. This technology could in theory be used for site-specific integration of a CAR or TCR. However, CRISPR/Cas9-mediated site-specific gene insertion is a homologous-directed DNA repair process with yet very low efficiency and potential off-target effects (Fu *et al*, 2013). Extensive genotyping is also needed to identify clones with a correct gene insertion.

Here, we present a new tool for T-cell engineering by combining NILV with S/MAR. The NILV-S/MAR-based method described herein yields reduced risk of genotoxicity, but retains high gene delivery efficiency and long-term transgene expression, similar to the conventional self-inactivating integrating LV vectors. This finding also paves way for safe engineering of CD34[+] hematopoietic

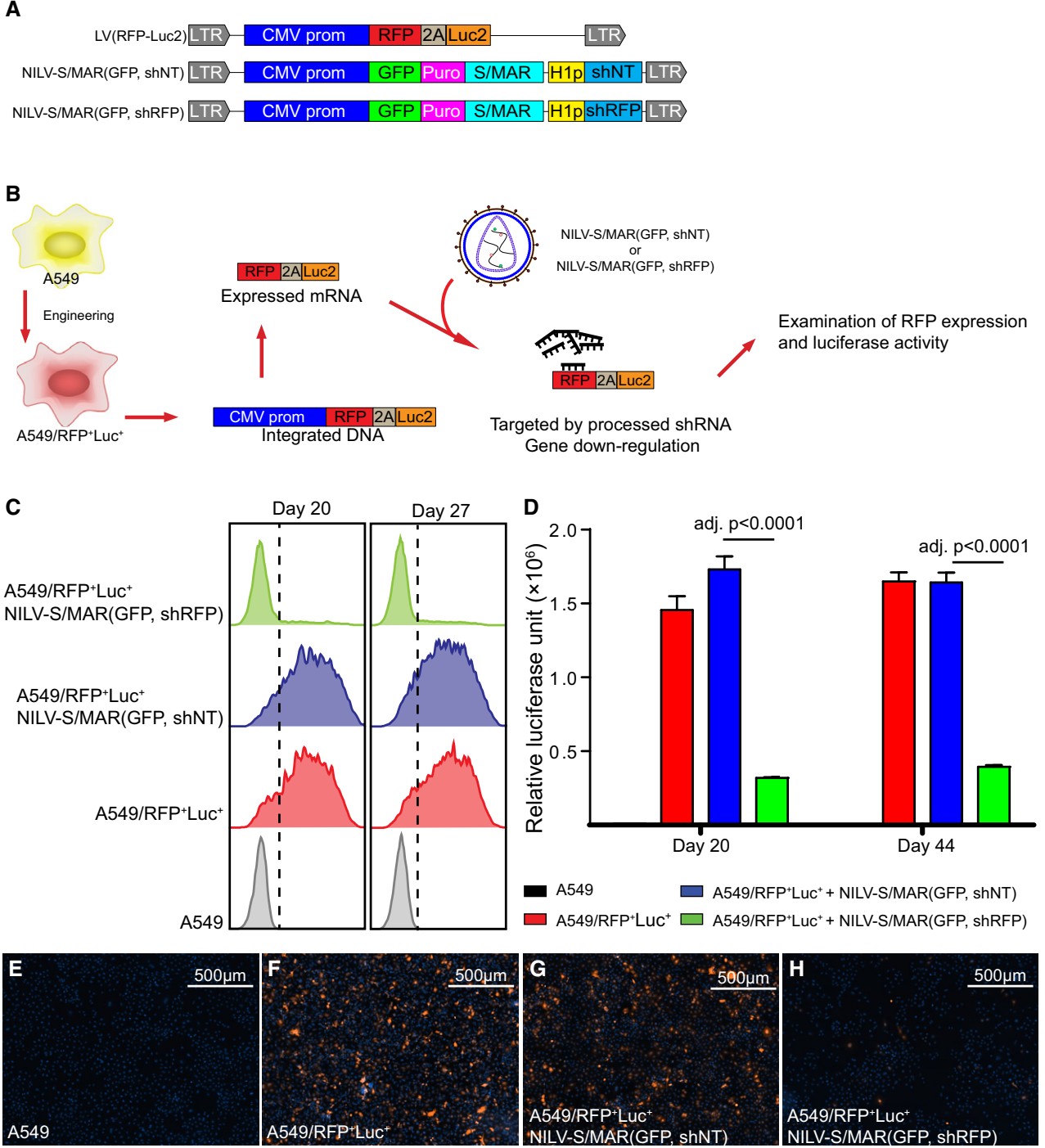

**Figure 2. NILV-S/MAR-mediated gene knockdown.**

A  Schematic drawing of the lentiviral vectors used. Abbreviations as in Fig 1; RFP: red fluorescent protein; Luc2: codon-optimized (for *Homo sapiens*) firefly luciferase; H1p: H1 RNA polymerase III promoter; shRFP: small hairpin RNA targeting the RFP transcript; shNT: small hairpin RNA with nontargeted control sequence.

B  The experimental setting for proof-of-concept experiment showing that a NILV-S/MAR-based vector can yield long-term gene silencing.

C  Flow cytometry histograms of RFP expression (*x*-axis) of A549/RFP+Luc+ cells at days 20 and 27 after NILV-S/MAR(GFP, shNT) or NILV-S/MAR(GFP, shRFP) transduction.

D  Luciferase activity of the A549/RFP+Luc+ cells at days 20 and 44 after NILV-S/MAR(GFP, shNT) or NILV-S/MAR(GFP, shRFP) transduction.

E–H  Representative fluorescence microscopy images show the expression of RFP from either NILV-S/MAR-transduced or untransduced A549/RFP+Luc+ cells with cell nuclei stained with Hoechst 33342 (blue). RFP signal (orange) was collected with a fixed exposure time (300 ms) for equal comparison.

Data information: Data are shown as mean ± SEM. Statistical analysis was performed using one-way ANOVA with Tukey's multiple comparisons test. Exact *P*-values are reported in Appendix Table S1.

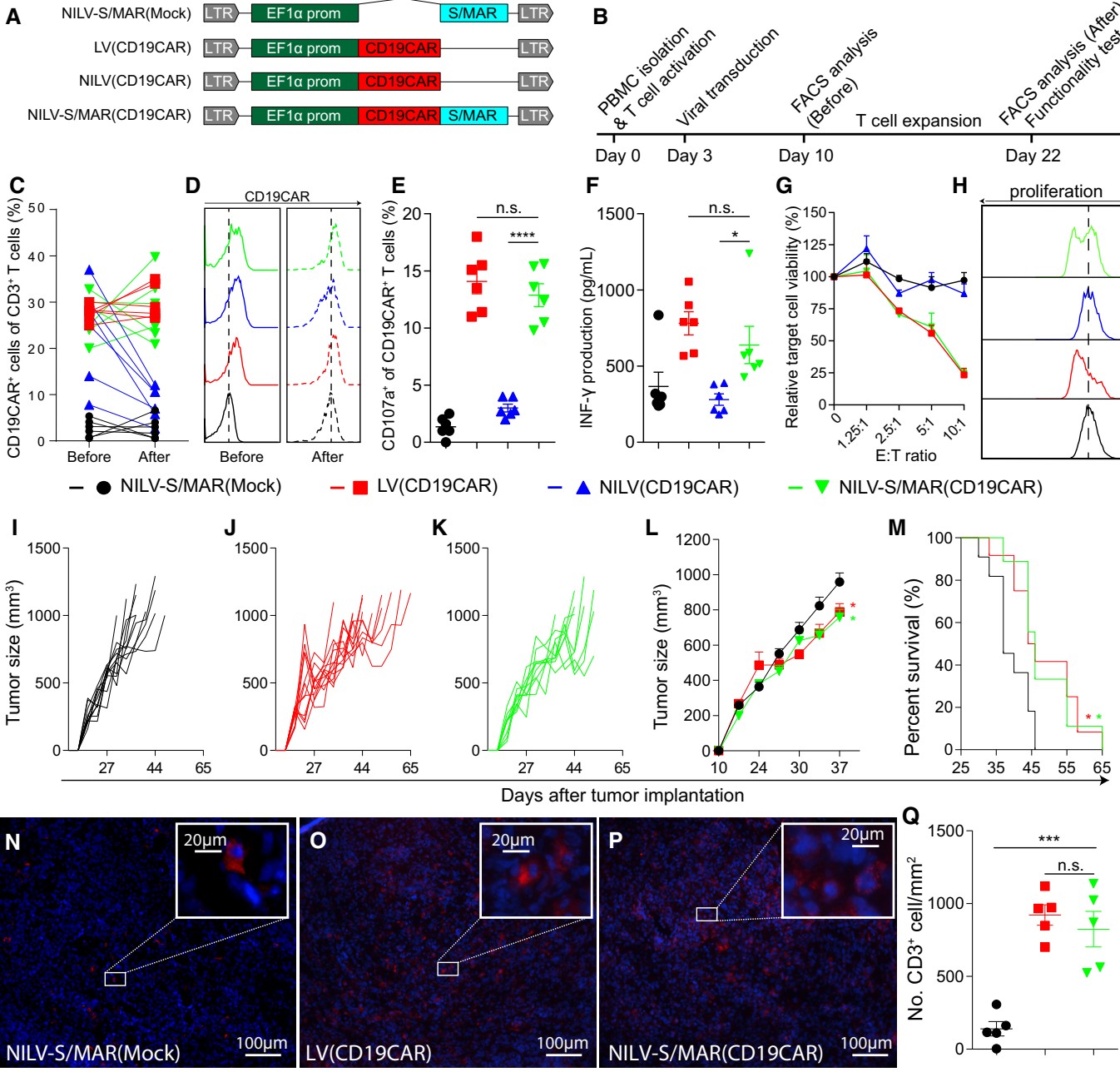

**Figure 3.**

stem cells (HSCs) where insertional mutagenesis has been observed when integrating RV and LV vectors were used for cell engineering (Ott *et al*, 2006; Hacein-Bey-Abina *et al*, 2008; Avedillo Diez *et al*, 2011) as well as for stem cells derived from other tissues, embryonic stem cells, and induced pluripotent stem cells.

Both for GFP engineering of A549 cells and CD19 CAR engineering of T cells, NILV-based vectors showed short-term expression, while NILV-S/MAR-based vectors secured long-term transgene expression. Furthermore, knockdown of gene expression in transduced A549 cells was equally efficient and stable. The results are in accordance with other reports and hypothesize that the S/MAR element facilitates DNA vector attachment to the nuclear matrix proteins during active gene transcription and then hitchhikes on the

cellular DNA replication machinery, which leads to self-replication of the extra-chromosomal DNA circles (Jenke *et al*, 2004b). Another advantage of S/MAR-based vectors is that they possess a lower immunological risk than an integrating viral vector (Jackson *et al*, 2006) due to the low copy-number persistence. In addition, S/MAR vectors are not subjected to epigenetic silencing (Jenke *et al*, 2004a), which can occur for integrating vectors through heterochromatinization. Thus, S/MAR vectors ensure maintenance of a transcriptionally active transgene.

In conclusion, we report a novel strategy for long-term T-cell engineering with reduced risks of genotoxicity and propose their use in immunotherapy of cancer. The engineering approach can possibly be extended to other applications such as hematopoietic

◄ **Figure 3.  Functionality of NILV-S/MAR-engineered CD19 CAR T cells.**

A    Schematic drawing of the lentiviral vectors used. Abbreviations as in Fig 1; EF1α prom: the human elongation factor-1α gene promoter; CD19CAR: chimeric antigen receptor (CAR) targeting the pan-B-cell marker CD19 (for detailed information, see Materials and Methods).

B    A schematic timeline of the experimental setting.

C    Percentage of CD19CAR expression on CD3[+] T cells from each independent donor.

D    Histograms of CD19 CAR expression on T cells form one representative donor before and after expansion.

E    CD107a expression on transduced T cells (CD3[+]) after exposure to CD19[+] target cells (Daudi) for 24 h ($n = 6$).

F    IFN-γ production in the supernatant was analyzed by ELISA after 24 h of exposure of transduced T cells to target cells ($n = 6$).

G    Relative viability of target cells (CD19[+] Daudi), as an indication of T-cell killing efficacy, was measured at different effector T-cell to target cell ratio (E:T ratio) and normalized to the signal from target cells alone ($n = 6$).

H    Transduced T cells were labeled with a fluorescent cell membrane dye and cocultured with CD19[+] Daudi cells for 4 days. Dye dilution upon T-cell division was monitored by flow cytometry and representative histograms are shown.

I–M  Mice were treated by intravenous injection of NILV-S/MAR(Mock)-, LV(CD19CAR)-, and NILV-S/MAR(CD19CAR)-transduced T cells ($8 \times 10^6$ cells/injection) at days 18, 21, and 24 after tumor implantation. Tumor growth was monitored by caliper measurements. (I–K) Tumor volume of individual mice in each treatment groups is presented. (L) The average tumor volume of each treatment group is presented and the values on day 37 were statistically compared ($n = 12$ in LV group, $n = 9$ in NILV-S/MAR group and $n = 11$ in mock control group). (M) The survival of mice is shown as Kaplan–Meier curves.

N–P  Representative images showing infiltration of adoptively transferred human T cells into tumors. T cells were visualized by staining for human CD3 (red) and nucleus (blue).

Q    Quantification of CD3[+] T cells from each treatment group is presented ($n = 5$).

Data information: Data are presented as mean ± SEM, and in some cases, results from each donor are presented. Statistical analysis was performed using one-way ANOVA followed by Tukey's multiple comparisons test, while the Kaplan–Meier survival curves were compared using log-rank test with Bonferroni correction. n.s. $P \geq 0.05$; *$P < 0.05$; ***$P < 0.001$; ****$P < 0.0001$. Exact $P$-values between all groups from each test are reported in Appendix Table S1.

stem cell engineering for gene therapy of genetic diseases. This strategy should be further evaluated in preclinical and clinical settings.

## Materials and Methods

### Cell isolation and culture

Peripheral blood mononuclear cells (PBMCs) were isolated from fresh buffy coats (Uppsala University Hospital, Uppsala, Sweden) by Ficoll-Paque separation (GE Healthcare Life Sciences, Uppsala, Sweden) and cultured in RPMI-1640 medium supplemented with 10% heat-inactivated fetal bovine serum (FBS), 2 mM L-glutamine, 10 mM HEPES, 1 mM sodium pyruvate, 100 IU penicillin/ml & 100 μg streptomycin/ml (1% PeSt), and 20 μM β-mercaptoethanol. The CD19-positive human B-cell Burkitt's lymphoma cell line Daudi, the CD19-positive human B-cell non-Hodgkin's lymphoma cell line Karpas 422, and the human lung carcinoma cell line A549 were purchased from ATCC (Manassas, VA) and cultured in RPMI-1640 supplemented with 10% heat-inactivated FBS (20% for Karpas 422), 1 mM sodium pyruvate, and 1% PeSt. The human embryonic kidney cell line 293T (ATCC) was cultured in DMEM supplemented with 10% heat-inactivated FBS, 1 mM sodium pyruvate, and 1% PeSt. All components and culture media were purchased from Invitrogen (Carlsbad, CA). All cells were cultured in humidified incubator with 5% $CO_2$ at 37°C and regularly screened for mycoplasma infection (MycoAlert™ Mycoplasma Detection Kit from Lonza).

### Lentivirus vector construction and production

The sequence encoding the CD19 CAR was designed *in silico* and synthesized (GenScript, Piscataway, NJ) to contain the FMC63 Fab fragment for antigen binding to CD19, the CD8 hinge and transmembrane region, the 4-1BB and CD3 zeta chain intracellular signaling domains (Milone *et al*, 2009). The synthesized fragment

was sub-cloned into a third-generation self-inactivating lentivirus vector under transcriptional control of the human elongation factor-1 alpha (EF1α) promoter. The 2.2 kb S/MAR fragment (Mielke *et al*, 1990) upstream of the human interferon-beta gene, the same sequence as reported in the pEPI-serial plasmids, was synthesized (GenScript) and sub-cloned directly downstream of the transgene coding sequence. Conventional LV encoding the either GFP or CD19 CAR was produced using the 3-helper plasmid system (Jin *et al*, 2014). NILV vectors were produced using the packaging plasmid p8.74DNW (Apolonia *et al*, 2007), which has three class I point mutations in the integrase coding sequence thus abolished the integration activity. Both LV and NILV vectors were pseudotyped with VSV-G and all the vectors were produced in 293T cells as described previously (Jin *et al*, 2014) and titrated using the Lentivirus qPCR Titer Kit (Applied Biological Materials, Richmond, BC, Canada).

### PCR assay to detect circular DNA after lentiviral LTR circularization

A549 cells ($1 \times 10^6$) were transduced with LV(GFP), NILV(GFP), NILV-S/MAR(GFP), and NILV-S/MAR[rev](GFP) at five infectious units (IFU) per cell. Untransduced cells served as negative control. The cells were puromycin-selected and cultured for 2–4 weeks before DNA was extracted using TRIzol™ reagent (Invitrogen) with standard protocol. A PCR was set up with primers pF_LTR and pR_LTR (Appendix Fig S2A) to detect formation of lentiviral LTR junctions. The PCR products were resolved on a 1% agarose gel.

### Southern blot

911 cells ($1 \times 10^6$) were transduced with either LV(GFP) or NILV-S/MAR(GFP) at 1 IFU/cell, and single cell clones were generated through limited dilution. Genomic DNA was isolated from established clones using the DNeasy Blood and Tissue kit (Qiagen, Hilden, Germany). The DNA was digested with EcoRI (a restriction enzyme that cuts once in the LV or NILV-S/MAR vectors) before

being resolved on agarose gel and transferred to a positively charged nylon membrane (Nytran SuPerCharge, Whatman, GE Healthcare Life Sciences) by downside alkaline transfer using TurboBlotter (GE Healthcare Life Sciences) according to the manufacturer's instruction. The membrane hybridization and detection were carried out essentially followed the instruction of the digoxigenin (DIG) DNA Labeling and Detection kit (Roche, Basel, Switzerland). In brief, the detection probe, against the whole viral genome, was prepared from a plasmid by digesting the DNA and labeling it with DIG-dUTP using Klenow fragment. Hybridization was carried out at 42°C for 16–18 h. The insoluble color development was generated using NBT/BCIP as substrate and the membrane was finally imaged using the GelDoc Imager (Bio-Rad, Hercules, CA).

### Padlock probe and rolling circle amplification assay to detect episomal circular DNA

A549 cells ($1 \times 10^6$) were transduced with LV(GFP), NILV(GFP), and NILV-S/MAR(GFP) at 5 IFU/cell. Untransduced cells served as negative control. The cells were puromycin-selected and cultured for 2–4 weeks before being transferred to staining slides (Superfrost Plus, ThermoFisher Scientific, Waltham, MA). The slides were harvested 1 day after seeding and assayed (Weibrecht *et al*, 2012) with slight modifications. Cells were fixed and permeabilized. They were then treated with nicking enzyme (Nt.BspQI, New England Biolabs, Ipswich, MA) followed by lambda exonuclease (Thermo-Fisher Scientific) to digest the nicked DNA strand (Appendix Fig S4). A specific hybridization padlock probe pPadlock (Invitrogen) was applied followed by ligation using T4 DNA ligase (Invitrogen). Rolling circle amplification (RCA) reaction was performed using phi29 polymerase (ThermoFisher Scientific) and the pRCA primer (Invitrogen) for signal enhancement, and the products were further hybridized with the Cy3-labeled detection oligo pDetect (Invitrogen). Nuclei were stained with Hoechst 33342 (Invitrogen). Slides were imaged in a fluorescence microscope AxioImager M2 (Zeiss, German).

### Validation of gene knockdown

A549 cells were transduced with a conventional integrating LV vector encoding red fluorescent protein (RFP) and codon-optimized firefly luciferase (Luc2). Positive and stable cells were FACS-sorted (FACSAria™, BD Biosciences, Stockholm, Sweden) and designated as A549/RFP$^+$Luc$^+$. Sorted A549/RFP$^+$Luc$^+$ cells were transduced with either NILV-S/MAR(GFP, shRFP) or NILV-S/MAR(GFP, shNT), puromycin-selected, and kept in culture for more than 1 month. The expression of RFP was monitored by flow cytometry (FACSCanto II, BD Biosciences) and the luciferase activity was measured in triplicates using Bright-Glo Kit (Promega, Stockholm, Sweden) with a luminometer (Wallac VICTOR$^2$, PerkinElmer, Turku, Finland) at indicated time points. RFP expression was also monitored by fluorescence microscopy. NILV-S/MAR-transduced cells were seeded on SuperFrost™ Plus slides (Thermo Scientific) and cultured overnight to allow them to attach. Cells were then fixed with 4% paraformaldehyde for 15 min at room temperature, stained with Hoechst 33342 (Invitrogen) and mounted with fluoromount G (Southern Biotech, Birmingham, AL) for imaging. Images were taken using a Zeiss AxioImager M2 (Zeiss, German) at BioVis

platform, Uppsala University. The RFP signal (orange) was collected with a fixed exposure time (300 ms) for comparison.

### T-cell engineering, expansion, and characterization

PBMCs ($5 \times 10^6$ cells) were activated for 3 days in culture medium with the anti-CD3 antibody OKT-3 (100 ng/ml, BioLegend, San Diego, CA) provided at day 1 and IL-2 (100 IU/ml, Proleukin, Novartis, Basel, Switzerland) provided at day 2. Activated T cells ($2 \times 10^6$ cells) were resuspended in $2 \times 10^7$ IFU concentrated viral vector, either LV(CD19CAR), NILV(CD19CAR), NILV-S/MAR(CD19CAR), or NILV-S/MAR(Mock) together with 5 µl of Sequa-brene™ (1 mg/ml, Sigma-Aldrich, St. Louis, MO) and IL-2 (100 IU) in a volume of 50 µl and incubated for 4 h. The T cells were re-transduced again the day after and cultured in 1 ml culture medium supplied with IL-2 (100 IU/ml) for 7 days. T cells were then expanded using the AEP expansion protocol as previously described (Jin *et al*, 2014).

Transduced T cells both before and after expansion were phenotypically characterized by surface marker staining and flow cytometry (FACSCanto II) analysis (FlowJo, Tree Star, Ashland, OR). CD19 CAR was detected by staining with an APC-conjugated rabbit anti-mouse Ig(H+L) Fab fragment (Invitrogen, Cat: A10539) together with an antibody mixture to distinguish CD3/CD4/CD8 (BD Biosciences, Ref: 342414). PD1 was stained with FITC-conjugated anti-human PD1 antibody (BioLegend, Cat: 329904). Tim3 was stained with PE-conjugated anti-human Tim3 antibody (BioLegend, Cat: 345006).

### IFN-γ secretion and CD107a expression

CD19 CAR T cells were rested for 3 days after expansion in low-dose IL-2 (20 IU/ml) before they were cocultured with Daudi target cells at ratio of 1:1 for 20–24 h. IFN-γ secretion from stimulated T cells was measured in supernatants by ELISA using an anti-IFN-γ antibody (Mabtech, Nacka Strand, Sweden) and the T cells were stained for expression of the degranulation marker CD107a using a FITC-labeled anti-CD107 antibody (BD Biosciences, Cat: 555800) and evaluated by flow cytometry.

### T-cell proliferation assay

CD19 CAR T cells were rested for 3 days after expansion in low-dose IL-2 (20 IU/ml) and then stained with CellTrace Violet dye (Invitrogen) and cultured with Daudi target cells at ratio of 1:1 for 4 days. The CD19 CAR T cells were then analyzed by flow cytometry to measure the dilution of the violet dye as an indication of cell proliferation.

### Cytotoxic assay

CD19 CAR T cells were rested for 3 days after expansion in low-dose IL-2 (20 IU/ml) and then cocultured with luciferase-tagged target cells (Daudi) at ratios ranging from 1.25:1 to 10:1 for 72 h in a total volume of 100 µl. The luciferase activity (as indicator of live target cell) was measured as mentioned previously (Jin *et al*, 2014). The relative cell viability was calculated by normalizing relative light unit (RLU) from samples against the RLU from target cells without T cells.

## Animal studies

The Uppsala and Stockholm Research Animal Ethics Committee has approved the experiments (C215/12, N164/15). Female, 4 weeks old, NOD-SCID mice were purchased from Janvier Lab (France) and allowed to adjust to the new environment at the University animal facility (Uppsala, Sweden) for 1 week, where they were kept in individually ventilated cages (max 6 mice per cage). Ten million Karpas 422 cells were mixed at a 1:1 volume ratio with Matrigel (BD Biosciences) and injected subcutaneously on day 0 in the right hind flank in a total volume of 100 μl. Mice were preconditioned by intraperitoneal injection of cyclophosphamide (Sendoxan, 50 mg/kg, Baxter Medical AB, Kista, Sweden) on day 15. Mice were randomized and treated with intravenous injections of virus-transduced and expanded CD19 CAR T cells (described above) on days 18, 21, and 24 at a dose of $8 \times 10^6$ T cells/injection in 200 μl. Nine to twelve mice per group were used. Tumor growth was monitored by caliper measurement and no blinding was done. The tumor volume was calculated using the ellipsoid volume formula: Volume = $\pi \times$ Length $\times$ Width$^2$/6. The mice were sacrificed either when the tumor volume exceeded 1 cm$^3$ or tumors became ulcerous. Animal house staff and veterinarians were consulted in this matter.

Three additional mice per group were sacrificed 5 days after the third T-cell injection (day 29). Tumors were excised and 0.8-μm frozen sections were prepared for immunofluorescence staining to detect T-cell infiltration. The sections were washed in 1xPBS with 0.5% Tween-20 (1×PBS-T) for 5 min and blocked with 4% BSA in 1×PBS-T for 1 h at RT. The sections were stained with a rabbit anti-human CD3 antibody (1:500) (Abcam, Cat: Ab109531) at 4°C overnight and washed twice with 1xPBS-T, followed by incubation with a secondary Alexa 568-conjugated goat anti-rabbit antibody (1:1,000) (Invitrogen, Cat: A-11036) for 1 h at RT in a dark humidified chamber. The sections were washed twice with 1xPBS-T, stained with Hoechst 33342 stain (Invitrogen) for 10 min at RT, and mounted with fluoromount G (Southern Biotech). The images were taken using Nikon's LED-based fluorescent microscopy by NIS-Elements (Bergman Labora AB, Danderyd, Sweden) and the number of CD3[+] T cells was numerated using the ImageJ software (National Institutes of Health).

## Data analysis

The sample size of animal study was determined according to our previous studies as well as published literatures with similar study design. No blinding was done. In most numerical data, value from each individual donor was presented together with the mean ± SEM. Statistical analysis was performed by GraphPad Prism v6.01 (La Jolla, CA, USA). Unless differently mentioned in the figure legends, means of multiple groups were compared using one-way ANOVA followed by Tukey's multiple comparisons test.

Expanded View for this article is available online.

## Acknowledgements

We would like to express our sincere thanks to Dr. Di Wu (Uppsala University and SciLifeLab Stockholm) for help with the padlock probe hybridization assay. The Swedish Children Cancer Foundation, the Swedish Cancer Society, the Swedish Research Council, and Gunnar Nilsson's Cancer Foundation supported this work. DY is the receiver of a postdoc fellowship grant from the Medical Faculty of Uppsala University (E and R Börjesson Fund, L Erikssons Fund and A Karlssons Fund). The funders had no role in study design, data collection and analysis, decision to publish, or preparation of the manuscript.

## The paper explained

### Problem

CAR T-cell therapy involves *ex vivo* engineering of the cancer patients' own T cells to express a CAR molecule and redirect the engineered T cell to recognize and attack tumor cells. T-cell engineering is mainly done with lentiviral (LV) or retroviral (RV) vectors, which leads the transgene (herein the CAR) to be permanently integrated into the cell genome for sustained expression. However, the semi-random integration of LV and RV raises concern that the engineered T cells can become tumorigenic. It has so far not been observed for T cells but observed in clinical trials wherein hematopoietic stem cells were engineered by RV vectors.

### Results

We present a novel episomal and long-term cell engineering technology. It is based on combining a non-integrating lentiviral (NILV) vector, wherein the integrase is mutated, and a scaffold/matrix attachment region (S/MAR) element, which anchors the episomal circularized DNA to the host cell genome, without integration, and hitchhikes on the cellular replication machinery. We illustrate the use of the NILV-S/MAR vector for expression of transgenes and down-regulation of target genes. We also affirm that NILV-S/MAR-engineered CAR T cells are as efficient as LV-engineered CAR T cells to kill tumor cells both *in vitro* and *in vivo*.

### Impact

The findings increase the safety level of T-cell engineering and can offer improved safety for engineering of other cell types where insertional mutagenesis caused by integrating RV/LV is a concern, including hematopoietic and tissue stem cells.

## Author contributions

CJ, ME, and DY conceived and designed the experiments; CJ, GF, MR, BN, and DY performed the experiments and analyzed the data; and CJ, ME, and DY wrote the paper. All authors discussed the results, read and approved the final version of the manuscript.

## Conflict of interest

The authors declare that they have no conflict of interest.

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
