## [Review Process File · EMBO Molecular Medicine]

Safe engineering of CAR T cells for adoptive cell therapy of cancer using long-term episomal gene transfer

Chuan Jin, Grammatiki Fotaki, Mohanraj Ramachandran, Berith Nilsson, Magnus Essand, Di Yu

Corresponding author: Magnus Essand, Uppsala University

Review timeline:

Submission date:	31 August 2014
Editorial Decision:	29 September 2014
Resubmission:	30 September 2016
Editorial Decision:	21 October 2016
Revision received:	11 March 2016
Editorial Decision:	23 March 2016
Revision received:	04 April 2016
Accepted:	08 April 2016

Transaction Report:

Editor: Roberto Buccione

1st Editorial Decision

29 September 2014

Thank you for the submission of your manuscript to EMBO Molecular Medicine. We have now heard back from the three Reviewers whom we asked to evaluate your manuscript.

As you will see, all Reviewers raise significant, fundamental and partially overlapping concerns that in aggregate, I am afraid, preclude publication of the manuscript in EMBO Molecular Medicine. I will not discuss each point in detail as they are clearly stated. There are, however, two crucial points that I wish to bring to your attention.

The main shared issues are the lack of a clear demonstration that the vectors do not integrate and missing proof of *in vivo* efficacy. As a result, the main interest of your manuscript, i.e. potential clinical application, remains to be demonstrated.

Given these fundamental concerns and the overall lack of enthusiasm by the Reviewers, I have no choice but to return the manuscript to you at this stage. In our assessment it is not realistic to expect to be able to address these issues experimentally in a reasonable time frame and to the satisfaction of the Reviewers.

I am sorry to have to disappoint you. I hope that the Reviewer comments will be helpful in your continued work in this area.

***** Reviewer's comments *****

Referee #1 (Remarks):

In this manuscript, the authors describe a novel non-integrating lentiviral vector that contains a scaffold matrix attachment region (S/MAR) that permits stable episomal replication, thus potentially avoiding the risk of insertional mutagenesis whilst maintaining long-term expression in transduced cells. Two cell systems are employed - A549 (NSCLC) cells and PBMC-derived T cells, with the majority of work with A549.

The use of non-viral S/MAR vectors has been described before. However, a recent publication (Verghese et al, reference 16) described an S/MAR-containing HIV-based vector. Thus, the novelty of these data are somewhat reduced. However, the potential to generate a non-integrating vector that allows long-term, high level CAR expression (or indeed any transgene) would have great clinical potential.

The experiments have well performed with adequate controls, although figure 1c lacks error bars, so it's not clear how many times experiments have been repeated to demonstrate reproducibility.

Major comments

1. In figure 1, the intensity (1b) and duration (1c) of GFP expression is significantly attenuated in both NILV-S/MAR vectors compared to the fully integrating vector. This is a potentially major problem. I also notice that all experiments stop at day 30 with GFP expression falling in the NILV-S/MAR populations: what happens beyond day 30?
2. The key to the success of these vectors is the absence of integration. Figure 1d confirms that the NILV-S/MAR-transduced populations only contain the 1-LTR product. However, this is not a convincing demonstration of absence of integration. Proving a negative is always difficult, but sequencing of 3 clones (presumably Sanger sequencing) does not demonstrate that there is no integration. The authors need to attempt to prove more robustly the absence of integration - Verghese et al used Southern blotting, but deep NGS would be preferable. Similarly, this should ideally be performed in the true target population (T cells) as well as A549.
3. Figure 2 is fine - it's essentially a re-statement of the principles of figure 1, with shRNA replacing GFP.
4. Figure 3 is critical. 3c and d demonstrate that CD19CAR expression can be maintained for 19 days during expansion - unlike GFP in A549 cells, the flow suggests that fluorescence intensity is similar in NILV-S/MAR and LV cells: it would be good to see MFI data as well as "% positive". 3e, f, g and g suggest that the NILV-S/MAR-transduced population degranulate, produce IFN- γ , lyse target cells and proliferate similarly to LV-transduced cells following expansion. The key issue is what would happen in vivo following delivery of such NILV-S/MAR-transduced CD19CAR T cells compared to LV-transduced cells - are they are effective? Do they persist to the same degree? I think that these experiments are crucial for longer-term clinical utility of these vectors.

Minor comment

1. Why does the integrating vector only generate 1-LTR circles (Fig 1d, lane 2)? Does this reflect aberrant NHEJ in A549 cells?

Referee #2 (Comments on Novelty/Model System):

The work submitted here is interesting but incomplete.

I would suggest that authors would further demonstrated the episomal status of their S/MAR-NILV vector as suggested in my review of the paper. This would add weight to the article and be a complementary demonstration that this S/MAR sequence keeps circles as extra chromosomal structures.

In addition I believe that an in vivo experiment is necessary to demonstrate the potential use of their CD19CAR gene transfer strategy to target myeloma cells as well as to establish the functionality of the episomal vector.

Referee #2 (Remarks):

The article, ' Long-term episomal gene transfer for safe engineering of T-cells for adoptive cell therapy of cancer" by Chuan Jin et al., describes the use of NILVs containing a scaffold matrix attachment region (S/MAR) for long term transgene expression in lymphocytes. The medical interest of this tool is demonstrated through the NILV-S/MAR mediated gene transfer of a CD19CAR to T-lymphocytes and the use of these engineered cells for B-cells depletion.

The goal of the work is clear, the article is well written and the proposed experiments are carefully performed with appropriate controls.

The main purpose of Chuan Jin et al. is to demonstrate that NILVs equipped with an S/MAR are as useful as an integrative LV but with the advantageous feature of remaining episomal. To engineer their NILV they use the S/MAR sequence of the human beta-interferon gene also used by Verghese, SC. et al. in a similar demonstration and published a few months ago in NAR. Here the authors also provide experimental clues of a possible clinical use of S/MAR-NILV.

Comments:

- In the first part of the study, the demonstration of the episomal status of S/MAR-NILV, which is central to the study, relies on the padlock probe assay and standard PCR of LTR junctions. In my opinion these experiments only weakly demonstrate that S/MAR-NILV remain extrachromosomal. a) Positive spots of hybridization are as numerous in integrating LV control as in S/MAR-NILV, which raises the question of why in the absence of the S/MAR sequence, so many episomes would remain in LV transduced cells and could suggest that the padlock probe may recognize some integrated forms of recombinated HIV circles with contiguous LTRs as I suppose it hybridizes to LTR junctions. In any case this ambiguous result requires additional experiments to demonstrate that the S/MAR does not increase NILV integration and/or do keeps the vector genome as an extrachromosomal circle.

- One possibility is to use real time PCR to quantify 1 and 2 LTR forms (Munir, S. et al. 2013 Retrovirology) in LV, NILV and S/MAR-NILV transduced cells, which would in addition be complementary to Verghese, SC. et al. demonstration.

- Another possibility would be to show that the circles can be released from their chromatin anchorage using protein/DNA dissociation with increasing salt concentration followed by HIV DNA measurements with QPCR in supernatants (see Astiazaran, P. et al. 2011 retrovirology).

Moreover it would be interesting to know if S/MAR-NILV episomes tend to be integrated within chromatin across cell divisions.

- Authors should mention in that first part that transgene expression is lower with NILV-S/MAR than with LV (Fig 1b) and discuss it. This should have implications for therapeutic strategies.

- In the second part of the paper a control LV-shRFP should have been used to evaluate the difference of efficiency in shutdown linked to the amount of shRFP expression (considering that LV express more transgene than S/MAR-NILV as shown in Fig. 1b).

-Why in Fig. 2g cells are not yellow and green in 2h should be mentioned. As GFP is a marker of shRFP expression this should be put in equation with the relatively low shut down of luciferase expression.

-The third part of the article shows the interest of a S/MAR-NILV vector for a biological purpose in cultured cells. It is shown that LV and S/MAR-NILV are equally efficient to transduce T-lymphocytes over a period of 3 weeks. In Figure 3d and contrary to what observed in Fig. 1b, CD19CAR expression appears similar for both vectors. This should be discussed.

Interestingly authors observed equivalent biological effects induced by engineered T-cells following LV or S/MAR-NILV transduction of a CD19CAR transgene. Upon CD19 B-cells recognition, T-cells-CD19CAR expressed CD107a, released IFN-g, proliferated and were able to kill CD19+ lymphocytes.

This is exciting. This work however would gain much quality if pushing the study a little further as for testing T-cells-CD19CAR engineered cells to target B-cells in a mouse model of myeloma or alternatively to target hybridoma cells in vivo and shut down (or reduce) antibodies production.

Minor comments:

-Somewhere the 3 mutations of integrase used should be mentioned. In Apolonia et al. several constructs with 3 mutations are used.

-Legends of Fig. 1, 2 and 3, mention the titers of vector used.

- Legend Fig. 1d, mention the time after transduction at which the PCR was done

Referee #3 (Comments on Novelty/Model System):

This paper shows that a S/MAR element can improve maintenance of a non-integrating lentiviral vector.

The observation has been published previously in NAR (ref17)

The experimental design is inadequate (see below)

The model system shows transduction of T cells with a chimeric antigen receptor, there is very little data on the functionality of these cells.

Referee #3 (Remarks):

This manuscript reports a potential maintenance of non-integrating lentiviral vectors by inclusion of a S/MAR element.

-This observation has been published previously (ref17).

-The experimental design does not support the authors claims. For example, in figure 1 the cells are transduced with various vectors then selected with puromycin. The authors provide no proof that the vectors have not integrated during the selection period, which would be likely as they behave very much as plasmids.

-The experimental data is very far from clinical application. If a chimeric antigen receptor is the chosen transgene then I would expect transduction of lymphocytes under standard clinical trial conditions, followed by demonstration that the transduced cells kill tumours in a xenograft experiment.

Referee #1 (Remarks):

In this manuscript, the authors describe a novel non-integrating lentiviral vector that contains a scaffold matrix attachment region (S/MAR) that permits stable episomal replication, thus potentially avoiding the risk of insertional mutagenesis whilst maintaining long-term expression in transduced cells. Two cell systems are employed - A549 (NSCLC) cells and PBMC-derived T cells, with the majority of work with A549.

The use of non-viral S/MAR vectors has been described before. However, a recent publication (Verghese et al, reference 16) described an S/MAR-containing HIV-based vector. Thus, the novelty of these data are somewhat reduced. However, the potential to generate a non-integrating vector that allows long-term, high level CAR expression (or indeed any transgene) would have great clinical potential.

The experiments have well performed with adequate controls, although figure 1c lacks error bars, so it's not clear how many times experiments have been repeated to demonstrate reproducibility.

Major comments

1. In figure 1, the intensity (1b) and duration (1c) of GFP expression is significantly attenuated in both NILV-S/MAR vectors compared to the fully integrating vector. This is a potentially major problem. I also notice that all experiments stop at day 30 with GFP expression falling in the NILV-S/MAR populations: what happens beyond day 30?

We agree with the reviewer's comment that it may pose a problem when using cell lines such as A459. We have complemented the experiment using a different cell line (911, immortalized human embryonic retinoblasts). Differently to the experiment with A549 cells (Figure 1), we did not use selection for 911 cells. The results are shown in Supplementary Figure S1A (% transgene expression) and S1C (expression level). The results are similar to the data presented for A549 cells in Figure 1c and 1b with attenuated expression.

We believe that the attenuated expression in A549 and 911 is due to the low copy number of S/MAR compared to copy numbers of transgenes obtained from integrating LV's. Others have reported the same observation when using cell lines [1,2]. As discussed below we do not see this for NILV-S/MAR-engineered human T cells.

We did not monitor the expression beyond day 38 but the expression levels for NILV-S/MAR-transduced 911 cells are stable from about day 18 to 38 (Supplementary Figure S1), which shows that a stable expression level and long-term expression have been achieved.

2. The key to the success of these vectors is the absence of integration. Figure 1d confirms that the NILV-S/MAR-transduced populations only contain the 1-LTR product. However, this is not a convincing demonstration of absence of integration. Proving a negative is always difficult, but sequencing of 3 clones (presumably Sanger sequencing) does not demonstrate that there is no integration. The authors need to attempt to prove more robustly the absence of integration - Verghese et al used Southern blotting, but deep NGS would be preferable. Similarly, this should ideally be performed in the true target population (T cells) as well as A549.

We agree that normal PCR and sequencing of clones may not be a fully convincing demonstration of non-integration.

As mentioned by the reviewer, Verghese et al. (Nucleic Acids Res. 2014 Apr;42(7):e53), which made a similar design to ours, used Southern blot to confirm non-interaction of vector. We have now complemented our assays using LAM-PCR to confirm that the vector stays episomal. This method was recently published to prove a non-integrational status of S/MAR vectors [3]. Our LAM-PCR results are now presented as Supplementary Figure S4. Integration was proven for LV and not detected for NILV-S/MAR LV.

We would like to stress that we have also used a Pad-Lock assay (Supplementary Figure S3) together with the LTR-junction PCR assay (Supplementary Figure S2) to show that the NILV-S/MAR vector does not integrate. We believe that three independent methods by us, together with results published by others using S/MAR-based vectors showing non-integration, should be sufficient proof that integration does not occur.

3. Figure 2 is fine - it's essentially a re-statement of the principles of figure 1, with shRNA replacing GFP.

Figure 2 shows that the technique cannot be used only for gene expression but also for target gene silencing. We believe that this is a useful application and that our manuscript adds value to the research field. Verghese et al did not identify or demonstrate this application.

4. Figure 3 is critical. 3c and d demonstrate that CD19CAR expression can be maintained for 19 days during expansion - unlike GFP in A549 cells, the flow suggests that fluorescence intensity is similar in NILV-S/MAR and LV cells: it would be good to see MFI data as well as "% positive". 3e, f, g and g suggest that the NILV-S/MAR-transduced population degranulate, produce IFN- γ , lyse target cells and proliferate similarly to LV-transduced cells following expansion. The key issue is what would happen *in vivo* following delivery of such NILV-S/MAR-transduced CD19CAR T cells compared to LV-transduced cells - are they effective? Do they persist to the same degree? I think that these experiments are crucial for longer-term clinical utility of these vectors.

We would like to emphasize that Figure 3c in fact shows percent positivity and Figure 3d shows MFI data.

We agree with the reviewer that the lack of *in vivo* efficacy data was the weakest point of our old manuscript. We have now performed *in vivo* experiment comparing NILV-S/MAR-transduced and LV-transduced CD19 CAR T cells. The results are presented as Figure 3i-m. Importantly, we see similar therapeutic effects for human CD19 CAR T cells obtained through transduction using the NILV-S/MAR-based and conventional LV-based vector. We hope that these results strengthen the manuscript enough to send it out to reviewers.

Minor comment

1. Why does the integrating vector only generate 1-LTR circles (Fig 1d, lane 2)? Does this reflect aberrant NHEJ in A549 cells?

We obtained the same results with A549 cells as Verghese et al did when they used HEK293 cells. It may reflect aberrant NHEJ in A549 (but then most likely also HEK293 cells), or a high rate of homologous recombination.

Referee #2 (Comments on Novelty/Model System):

The work submitted here is interesting but incomplete.

I would suggest that authors would further demonstrated the episomal status of their S/MAR-NILV vector as suggested in my review of the paper. This would add weight to the article and be a complementary demonstration that this S/MAR sequence keeps circles as extra chromosomal structures. In addition I believe that an in vivo experiment is necessary to demonstrate the potential use of their CD19CAR gene transfer strategy to target myeloma cells as well as to establish the functionality of the episomal vector.

Referee #2 (Remarks):

The article, "Long-term episomal gene transfer for safe engineering of T-cells for adoptive cell therapy of cancer" by Chuan Jin et al., describes the use of NILVs containing a scaffold matrix attachment region (S/MAR) for long term transgene expression in lymphocytes. The medical interest of this tool is demonstrated through the NILV-S/MAR mediated gene transfer of a CD19CAR to T-lymphocytes and the use of these engineered cells for B-cells depletion.

The goal of the work is clear, the article is well written and the proposed experiments are carefully performed with appropriate controls.

The main purpose of Chuan Jin et al. is to demonstrate that NILVs equipped with an S/MAR are as useful as an integrative LV but with the advantageous feature of remaining episomal. To engineer their NILV they use the S/MAR sequence of the human beta-interferon gene also used by Verghese, SC. et al. in a similar demonstration and published a few months ago in NAR. Here the authors also provide experimental clues of a possible clinical use of S/MAR-NILV.

Comments:

- In the first part of the study, the demonstration of the episomal status of S/MAR-NILV, which is central to the study, relies on the padlock probe assay and standard PCR of LTR junctions. In my opinion these experiments only weakly demonstrate that S/MAR-NILV remain extrachromosomal.

a) Positive spots of hybridization are as numerous in integrating LV control as in S/MAR-NILV, which raises the question of why in the absence of the S/MAR sequence, so many episomes would remain in LV transduced cells and could suggest that the padlock probe may recognize some integrated forms of recombinated HIV circles with contiguous LTRs as I suppose it hybridizes to LTR junctions. In any case this ambiguous result requires additional experiments to demonstrate that the S/MAR does not increase NILV integration and/or do keeps the vector genome as an extrachromosomal circle.

- One possibility is to use real time PCR to quantify 1 and 2 LTR forms (Munir, S. et al. 2013 Retrovirology) in LV, NILV and S/MAR-NILV transduced cells, which would in addition be complementary to Verghese, SC. et al. demonstration.

- Another possibility would be to show that the circles can be released from their chromatin anchorage using protein/DNA dissociation with increasing salt concentration followed by HIV DNA measurements with QPCR in supernatants (see Astiazaran, P. et al. 2011 retrovirology).

We agree with the reviewer that the results may not provide enough proof of non-integration and we thank the reviewer for the suggestions. In addition, a LAM-PCR method was recently published to prove

the non-integrational status of S/MAR vectors [3]. We have now used LAM-PCR and own results (Supplementary Figure S4) show that the NILV-S/MAR vector does not integrate while the LV vector does.

We have used three independent methods to demonstrate that the NILV-S/MAR vector stays episomal and does not integrate in the host cell genome. These are:

- LTR-junction PCR, Supplementary Figure S2
- Pad-Lock assay, Supplementary Figure S3
- LAM-PCR assay, Supplementary Figure S4

We hope that our data together with published data by others, showing that plasmids with S/MAR elements do not integrate (1, 2) and the Southern blot data by Verghese et al. (Nucleic Acids Res. 2014 Apr;42(7):e53) showing that a non-integrating lentiviral vector with S/MAR does not integrate, should be sufficient proof.

Moreover it would be interesting to know if S/MAR-NILV episomes tend to be integrated within chromatin across cell divisions.

- Authors should mention in that first part that transgene expression is lower with NILV-S/MAR than with LV (Fig 1b) and discuss it. This should have implications for therapeutic strategies.

We agree and reviewer 1 also brought up this point. We have now added a discussion about this in the new version of the manuscript.

- In the second part of the paper a control LV-shRFP should have been used to evaluate the difference of efficiency in shutdown linked to the amount of shRFP expression (considering that LV express more transgene than S/MAR-NILV as shown in Fig. 1b).

We believe that lentiviral vectors carrying shRNA have already been proven to be the most efficient tool to achieve stable gene silencing in cells. It is a very widely used approach and a comparison will not add much new information. The aim of the study was to show proof-of-concept feasibility on using the NILV-S/MAR vector for gene down regulation and the important point is demonstrating specificity by comparing NILV-S/MAR with shNT (not targeted) and NILV-S/MAR with shRFP (specifically targeted). We reached approximately 75% (shRFP/shNT) gene knockdown efficacy, which is similar to what have been published using either siRNA or shRNA. We therefore believe the efficacy is well demonstrated.

-Why in Fig. 2g cells are not yellow and green in 2h should be mentioned. As GFP is a marker of shRFP expression this should be put in equation with the relatively low shut down of luciferase expression.

We thank the reviewer for commenting on this. The comment makes perfect sense. When taking Figures 2g and 2h we did not include the green channel in the microscope. We have now added pictures including the GFP channel, as supplementary Figure S5 (and below). When the A549/RFP⁺Luc⁺ cells were transduced with NILV-S/MAR (GFP, shNT) the cells become green but they also keep the red color from the earlier LV(RFP-Luc2)-transduction. An overlay picture makes the cells look yellow. This does not happen for NILV-S/MAR (GFP, shRFP)-transduced A549/RFP⁺Luc⁺ cells where the red color disappears due to rather efficient knock-down of RFP. The cells are therefore green.

As mentioned above, the shRFP gives ~75% shut down of Luc expression, which is in the normal range of shRNA efficacy. We therefore do not agree that the shut down efficacy is relatively low.

A549/RFP+Luc+
NILV-S/MAR(GFP, shNT)

A549/RFP+Luc+
NILV-S/MAR(GFP, shRFP)

-The third part of the article shows the interest of a S/MAR-NILV vector for a biological purpose in cultured cells. It is shown that LV and S/MAR-NILV are equally efficient to transduce T-lymphocytes over a period of 3 weeks. In Figure 3d and contrary to what observed in Fig. 1b, CD19CAR expression appears similar for both vectors. This should be discussed.

Interestingly authors observed equivalent biological effects induced by engineered T-cells following LV or S/MAR-NILV transduction of a CD19CAR transgene. Upon CD19 B-cells recognition, T-cells-CD19CAR expressed CD107a, released IFN-g, proliferated and was able to kill CD19+ lymphocytes.

This is exciting. This work however would gain much quality if pushing the study a little further as for testing T-cells-CD19CAR engineered cells to target B-cells in a mouse model of myeloma or alternatively to target hybridoma cells *in vivo* and shut down (or reduce) antibodies production.

We agree, after 3 weeks of T cell expansion, LV-transduced and NILV-S/MAR-transduced T cells show equally efficient transgene expression (Figure 3d). This was not the case for LV-transduced and NILV-S/MAR-transduced A549 cells (Figure 1b) or 911 cells (Suppl Fig S1). We do not know exactly why there is a discrepancy but we speculate that a possible reason for this is that the stability of S/MAR-based gene expression depends on the cell division rate. During expansion, T cells proliferate extensively when cultured together with allogeneic feeder cells and IL-2. The A549 and 911 cell lines also proliferate but not at the same rate.

We agree that lack of *in vivo* efficacy data was a weak part in the previous version of the manuscript. We have now performed *in vivo* experiment with CD19 CAR T cells (human T cells) for treatment of human CD19⁺ B cell tumors in a xenograft mouse model. The results are shown as Figure 3i-m. We observe comparable efficacy using NILV-S/MAR and conventional LV vectors. We hope that these results strengthen the manuscript enough to send it out for revision.

Minor comments:

-Somewhere the 3 mutations of integrase used should be mentioned. In Apolonia et al. several constructs with 3 mutations are used.

-Legends of Fig.1, 2 and 3, mention the titers of vector used.

- Legend Fig. 1d, mention the time after transduction at which the PCR was done

We thank the reviewer for the comments, we have now changed text accordingly.

R#3 did not give any more comments other than mentioned by R#1 and R#2

References

1. Jenke BH, Fetzer CP, Stehle IM, Jonsson F, Fackelmayer FO, et al. (2002) An episomally replicating vector binds to the nuclear matrix protein SAF-A in vivo. EMBO Rep 3: 349-354.
2. Jenke AC, Stehle IM, Herrmann F, Eisenberger T, Baiker A, et al. (2004) Nuclear scaffold/matrix attached region modules linked to a transcription unit are sufficient for replication and maintenance of a mammalian episome. Proc Natl Acad Sci U S A 101: 11322-11327.
3. Voigtlander R, Haase R, Muck-Hausl M, Zhang W, Boehme P, et al. (2013) A Novel Adenoviral Hybrid-vector System Carrying a Plasmid Replicon for Safe and Efficient Cell and Gene Therapeutic Applications. Mol Ther Nucleic Acids 2: e83.

Thank you for the submission of your manuscript to EMBO Molecular Medicine. We have now heard back from the Reviewers whom we asked to evaluate your manuscript.

As you will see, two of the previous Reviewers and a new one evaluated your manuscript, which was considered as a de novo submission. The remaining concerns expressed by the first two and the ones from the third remain fundamental and important. I will highlight below the main issues.

Reviewer 1 notes the critical lack of experimental detail and statistical treatment that impairs proper evaluation of the data.

Reviewer 2 also laments the poor presentation and insufficient detail for the LAM-PCR experimentation and disagrees that lack of insertion is demonstrated. This reviewer also notes that the in vivo data suffers from the low numbers of animals and, again lack of proper statistical treatment and inappropriate presentation. I agree that the in vivo data are very important but not completely convincing.

Reviewer 3 also agrees that the approaches taken are not sufficient to address the issue of integration, although s/he is somewhat less preoccupied of the importance of this. S/he also suggests an alternative approach that could possibly solve the issue. Finally, this Reviewer also suggests that a more balanced account of the available knowledge should be given.

In conclusion, while publication of the paper cannot be considered at this stage, we have decided to give you the opportunity to address the above concerns.

We are thus prepared to consider a substantially revised submission, with the understanding that the Reviewers' concerns must be addressed with additional experimental data where appropriate and that acceptance of the manuscript will entail a second round of review.

The overall aim is to significantly upgrade the clinical relevance and usefulness of the dataset, which of course is of paramount importance for our title.

It is important that you consider that it is EMBO Molecular Medicine policy to allow a single round of revision only and that, therefore, acceptance or rejection of the manuscript will depend on the completeness of your responses included in the next, final version of the manuscript.

As you know, EMBO Molecular Medicine has a "scooping protection" policy, whereby similar findings that are published by others during review or revision are not a criterion for rejection. However, I do ask you to get in touch with us after three months if you have not completed your revision, to update us on the status. Please also contact us as soon as possible if similar work is published elsewhere.

Finally, please note that EMBO Molecular Medicine now requires a complete author checklist (<http://embomolmed.embopress.org/authorguide#editorial3>) to be submitted with all revised manuscripts. Provision of the author checklist is mandatory at revision stage; The checklist is designed to enhance and standardize reporting of key information in research papers and to support reanalysis and repetition of experiments by the community. The list covers key information for figure panels and captions and focuses on statistics, the reporting of reagents, animal models and

human subject-derived data, as well as guidance to optimise data accessibility.

I look forward to seeing a revised form of your manuscript as soon as possible.

***** Reviewer's comments *****

Referee #1 (Comments on Novelty/Model System):

Similar experiments have been published recently either concerning the stability of NILV in dividing cells if containing a S/MAR element (Verghese SC, *Nucleic Acids Res.* 2014 Apr;42(7)) or the CD19 CAR gene transfer in T-cells in clinics (Sommermeyer D, *Leukemia.* 2015 Sep 15). The novelty here combines the 2 aspects, eventually providing a safer approach but no conceptual innovation.

Referee #1 (Remarks):

Reviewer 2:

Authors of the article "Long-term episomal gene transfer for safe engineering of T-cells for adoptive cell therapy of cancer", have addressed major comments of a first review of their manuscript providing new evidences obtained with additional experiments.

1) Regarding the weak or ambiguous proofs of the episomal status of their vector NILV-S/MAR as provided with regular PCR and padlock probe / rolling circle amplification, authors have now added a LAM-PCR experiment showing that only LV (integrase competent vectors) are integrated. However, very little information is given about how LAM-PCR was realized (only a reference). Most importantly in this new version of the paper, authors forgot to mention the type of transduced cells they choose to analyze and the time after transduction at which DNA was collected to perform LAM-PCR. This result would be most valuable if realized at latest time points on circulating T-cells of the in vivo experiment. Moreover, authors show only a single band when LAM-PCR on polyclonal transduced cells, should give several bands of variable size.

2) Authors have now completed their study with a requested in vivo experiment that seem to demonstrate that transduced T-cells expressing CD19 CAR are able to contain tumor growth and extend animal's survival. This is exciting but requires further statistical arguments to determine whether these trends are actually significant as stated in the core of results.

Referee #2 (Remarks):

The authors have submitted a revised version of their 2014 manuscript with new data and answers to reviewers' comments.

In the previous manuscript, there were two areas in particular that this reviewer felt needed to be addressed. The first was proof of non-integration and the second was demonstration of in vivo efficacy.

As stated in the original review, I acknowledge that proving a negative is difficult, and the authors present data beyond their original conventional PCR and Sanger sequencing. A LAM-PCR experiment is presented in supplementary figure 4. However, this is somewhat confusing and not very convincing. The figure legend states that "The band (red arrow) at size 225 bp corresponds to the internal control." The red arrow actually points to a thin band at c.100bp, the likely insertion band in LV-transduced cells. However, there also appears to be a band at c.120bp in the non-integrating NILV lane, which is neither mentioned nor explained.

Supplementary figure 2 shows some sequencing of PCR products, again suggesting the presence of a 1-LTR product. However, this does not truly show that there is not insertion - it shows that there is demonstrable 1-LTR PCR product (akin to figure 1d). The Pad-lock assay in figure 1e - g aims to show the persistence of circularized DNA in the NILV-S/MAR-transduced cells (1g), but not the NILV cells (1f); I agree that the difference between these two populations is clear. However, the presence of marked fluorescence signals in the LV-transduced cells suggests that this assay does not truly demonstrate lack of insertion.

In terms of in vivo efficacy, Karpas 422 tumours were injected s/c in NOD-SCID mice. Following cyclophosphamide on day 5, mice received IP T cells on days 7, 9 and 11. Unfortunately, group sizes were very small (n = 3 or 4) for the efficacy part of this experiment. Tumours in the mock-treated mice continued to grow exponentially. In both the LV and NILV-S/MAR-treated cohorts, two tumours continued to grow exponentially, one grew a little slower and one did not grow. No statistics are presented as to mean tumour volumes. Kaplan Meier survival curves (3j) are not really appropriate for s/c experiments. These results are interesting - one mouse from four did not grow following treatment with LV or NILV-S/MAR - but the results are suggestive rather than being truly convincing.

Referee #3 (Comments on Novelty/Model System):

The authors have performed several experiments that were well performed and with adequate controls. Even if a lentiviral vector with s/MAR-containing sequences has been described the Authors showed that this system could have some potential for anticancer therapy in vivo.

Referee #3 (Remarks):

In the manuscript from Chuan Jin et al., the Authors describe the development of a vector platform based on integrase defective lentiviral vectors (IDLVs) containing a scaffold matrix attachment region (S/MAR) to enable episomal replication and long term transgene expression in proliferating cells. The possibility to achieve stable genetic engineering without integrating in the target cell genome would avoid the risks of insertional mutagenesis. The Authors convincingly show that the integration proficient LV and the IDLV-S/MAR designs allowed stable transgene expression in ex-vivo transduced cells (although with different efficiencies) while the IDLV without the S/MAR sequences failed to provide stable transgene expression. The Authors show that IDLV-S/MAR based vectors can be used to achieve stable knockdown of gene expression in cell lines. Finally, the Authors engineered T cells with a IDLV-S/MAR based vector encoding a 2nd generation CAR against the CD19 molecule and achieved stable CAR expression and biological effects in vitro and in vivo equivalent to those achieved with T-cells engineered with an integration proficient LV. In summary the Manuscript shows that this platform is an interesting alternative platform for the safe genetic engineering of somatic cells in basic and translational applications.

Comments

The manuscript is interesting and, in general, the data is convincing. Moreover, the results obtained *in vivo* by the CD19 CAR-T cells are very encouraging.

The following issue however should be addressed:

Like for Adeno Associated Vectors (AAVs) and plasmids, it is possible that also the IDLV-S/MAR vectors integrate in the cell genome at low frequency using the cellular DNA repair machinery pathways. It will be interesting to estimate the levels (if any) of unwanted integration by this vector platform. The Authors used the specific PCR, lock-pad approach, and Linear Amplification Mediated (LAM) PCR to confirm the episomal status of the IDLV-S/MAR vector and to exclude genomic integration. Unfortunately none of these approaches fully address these issues.

a) Vector specific PCRs designed to amplify one or two LTR sequences present in the circularized of episomal forms cannot be used to exclude integration of vectors that integrate by DNA repair based mechanisms because the circularized DNA forms can integrate and still retain the two or single LTR sequences.

b) The specificity of the lock-pad assay is questionable. While the cells treated with the IDLV without S/MAR sequences do not show any signal in their nuclei (as expected), the integration proficient LV shows staining levels similar to those found in the nuclei of IDLV-S/MAR cells after puromycin selection. These results suggest that the signal seen in cells treated with the integrase proficient LV arise from integrated forms since its episomal forms, lacking the S/MAR sequences like the IDLV control, should be lost when cells proliferate. Therefore it is not possible to formally conclude that the signals seen in the IDLV-S/MAR treated cells are specific for episomal forms.

c) The patterns of LAM PCR products obtained from the different conditions (LV, IDLV-S/MAR and IDLV) showed a smear of bands only when cells were treated with the integrase proficient LV. Unfortunately this technique is well suited to retrieve vector/cellular genomic junctions of vectors that integrate precisely like those generated by the active LV integrase. On the contrary, LAM PCR is inefficient in retrieving the vector/genome junctions of the integrated AAVs or plasmids that do not integrate in a precise manner and thus do not create specific vector ends like those generated by the LV integrase. Therefore it is not a surprise that the LAM PCR bands in the agarose gel electrophoresis appear only when the cells are treated with the integration proficient LV.

To measure the levels of unwanted integration (if detectable) the Authors could use a Southern blot strategy in which the DNA from cells transduced with the different vectors (LV, IDLV-S/MAR and IDLV) and kept in culture without selective pressure is digested with a restriction enzyme that cut only once in the vector genome and probed for vector sequences. Digested circular episomal vector forms will be linearized and display a specific size while the integrated vector forms will display a different size depending on the position of the restriction site on the host cell and vector genomes.

Minor issue

The Authors fail to mention that the novel gene therapy vectors have an advanced design with self-inactivating LTRs and have an improved safety profile with respect those used previously. This of course does not mean that non integrating vectors are not a significant safety improvement in the gene therapy field. However, given the wide use of integrating LV in gene therapy it would be fair to provide a more balanced literature background.

***** Reviewer's comments *****

Referee #1 (Comments on Novelty/Model System):

Similar experiments have been published recently either concerning the stability of NILV in dividing cells if containing a S/MAR element (Vergheze SC, Nucleic Acids Res. 2014 Apr;42(7)) or the CD19 CAR gene transfer in T-cells in clinics (Sommermeyer D, Leukemia. 2015 Sep 15). The novelty here combines the 2 aspects, eventually providing a safer approach but no conceptual innovation.

We are fully aware of this fact. We submitted the manuscript to EMBO Molecular Medicine the first time in 2014 at the same time as the Vergheze et al., publication appeared. We have therefore refocused more on the application and hope that by doing so we can obtain sufficient interest to publish our paper in EMBO Molecular Medicine. There are many publications on CD19 CAR T cell therapy of leukemia and lymphoma in the clinic and it is fair to say that this therapy is about to be established. We have also performed a clinical CD19 CAR T cell study with 15 patients that we are to submit this spring. We believe, that the fact that the NILV-S/MAR vector is as efficient as the conventional LV vector in CD19 CAR transgene delivery makes our vector an attractive option for further studies.

Referee #1 (Remarks):

Reviewer 2:

Authors of the article "Long-term episomal gene transfer for safe engineering of T-cells for adoptive cell therapy of cancer", have addressed major comments of a first review of their manuscript providing new evidences obtained with additional experiments.

1) Regarding the weak or ambiguous proofs of the episomal status of their vector NILV-S/MAR as provided with regular PCR and padlock probe / rolling circle amplification, authors have now added a LAM-PCR experiment showing that only LV (integrase competent vectors) are integrated. However, very little information is given about how LAM-PCR was realized (only a reference). Most importantly in this new version of the paper, authors forgot to mention the type of transduced cells they choose to analyze and the time after transduction at which DNA was collected to perform LAM-PCR. This result would be most valuable if realized at latest time points on circulating T-cells of the in vivo experiment. Moreover, authors show only a single band when LAM-PCR on polyclonal transduced cells, should give several bands of variable size.

We thank the reviewer for observing the lack of information and apologize for the same. The LAM-PCR was performed on virus-transduced A549 cells and analyzed 4 weeks after transduction. This is now mentioned in the new text of the supplementary figure legend. The cited reference is a "Step-by-step" protocol, so we decide not to repeat the method in our text.

It is correct that the LAM-PCR gives several bands of variable size. However, with sizes around 100 bp, which is the case in our experiment, differences of a few bps cannot be distinguished in a normal agarose gel. Instead a somewhat wider band appears. Moreover, even for monoclonal transduced cells, due to the possibility of multiple insertions into a single cell, LAM-PCR result could give several bands of variable size instead of a single clear band also

in a single cell clone.

2) Authors have now completed their study with a requested *in vivo* experiment that seem to demonstrate that transduced T-cells expressing CD19 CAR are able to contain tumor growth and extend animal's survival. This is exciting but requires further statistical arguments to determine whether these trends are actually significant as stated in the core of results.

We agree with the reviewer and have therefore repeated the *in vivo* experiment with 12 mice per group plus 3 extra mice per group to study infiltration of CD19 CAR T cells into tumors. The results are similar to the data we obtained with only a few mice before. However, we now have a large enough data set to perform a statistical analysis, which is shown in Figure 3i-m. We show a significant difference between treated or non-treated mice. T cells engineered with our NILV-S/MAR vector did equally well as T cells engineered with the integrating LV.

T cell infiltration data are shown in Figure 3n-q. We find equal amount of human T cells in mice infused with LV-engineered and NILV-S/MAR-engineered CD19 CAR T cells.

Referee #2 (Remarks):

The authors have submitted a revised version of their 2014 manuscript with new data and answers to reviewers' comments.

In the previous manuscript, there were two areas in particular that this reviewer felt needed to be addressed. The first was proof of non-integration and the second was demonstration of *in vivo* efficacy.

As stated in the original review, I acknowledge that proving a negative is difficult, and the authors present data beyond their original conventional PCR and Sanger sequencing. A LAM-PCR experiment is presented in supplementary figure 4. However, this is somewhat confusing and not very convincing. The figure legend states that "The band (red arrow) at size 225 bp corresponds to the internal control." The red arrow actually points to a thin band at c.100bp, the likely insertion band in LV-transduced cells. However, there also appears to be a band at c.120bp in the non-integrating NILV lane, which is neither mentioned nor explained.

Supplementary figure 2 shows some sequencing of PCR products, again suggesting the presence of a 1-LTR product. However, this does not truly show that there is not insertion - it shows that there is demonstrable 1-LTR PCR product (akin to figure 1d).

We agree with the reviewer that the experiments were described too hasty. For the LAM-PCR experiment, the "red arrow" is a mistake in the text. The correct text should be "The band at size 225 bp corresponds to the internal control (that the viral DNA sequence is present in the sample). The narrow bands at approximately 100 bp (red arrow) correspond to insertion events."

The 120 bp band in the NILV(GFP) lane should be unspecific background product since the same faint band is visible in untransduced cells. Importantly, neither NILV nor untransduced cells have the internal control band (225bp), suggesting that there is no more detectable viral DNA left in these cells. This assay verifies insertional events of LV but does not show

insertional events for NILV-S/MAR, or as we state in the text, if so the insertional events are below detection level.

In order to further confirm that NILV-S/MAR do not cause genomic insertion and is maintained as DNA circles, we performed Southern Blot from single cell clones (Figure 1e). One clear 6.8 kb band can be detected from 2 different clones of NILV-S/MAR transduced cells, the same size as positive control (linearized plasmid). We believe that this proves that the NILV-S/MAR virus vector is maintained as circle DNA. This band is not seen for single cell clones from LV-transduced cells. The Southern Blot shows a different pattern for LV-transduced cells, indicating insertional events (which we know is the case).

The Pad-lock assay in figure 1e - g aims to show the persistence of circularized DNA in the NILV-S/MAR-transduced cells (1g), but not the NILV cells (1f); I agree that the difference between these two populations is clear. However, the presence of marked fluorescence signals in the LV-transduced cells suggests that this assay does not truly demonstrate lack of insertion.

We agree with reviewer's comment. The Pad-lock assay can only prove the existence of circularized DNA in NILV-S/MAR transduced cells. We apologize if our text indicated otherwise. The text in the new version of the manuscript has been changed and does not.

Assays looking at integration events are the LAM-PCR and Southern Blot. The LAM-PCR shows that if integrase-mediated integration occurs, it is at least below detection level and this is also the phrase we now use throughout the paper. We have also performed Southern Blot to prove that NILV-S/MAR is maintained as circularized DNA forms with possible integration events below detection level (Figure 1e). Both for the LAM-PCR and Southern Blot assays are we able to detect integration for LV-transduced cells.

In terms of in vivo efficacy, Karpas 422 tumours were injected s/c in NOD-SCID mice. Following cyclophosphamide on day 5, mice received IP T cells on days 7, 9 and 11. Unfortunately, group sizes were very small (n = 3 or 4) for the efficacy part of this experiment. Tumours in the mock-treated mice continued to grow exponentially. In both the LV and NILV-S/MAR-treated cohorts, two tumours continued to grow exponentially, one grew a little slower and one did not grow. No statistics are presented as to mean tumour volumes. Kaplan Meier survival curves (3j) are not really appropriate for s/c experiments. These results are interesting - one mouse from four did not grow following treatment with LV or NILV-S/MAR - but the results are suggestive rather than being truly convincing.

We agree with the reviewer and have therefore repeated the *in vivo* experiment with 12 mice per group plus 3 extra mice per group to study infiltration of CD19 CAR T cells into tumors. The results are similar to the data we obtained with only a few mice before. However, we now have a large enough data set to perform a statistical analysis, which is shown in Figure 3i-m. We show a significant difference between treated or non-treated mice. T cells engineered with our NILV-S/MAR vector did equally well as T cells engineered with the integrating LV.

T cell infiltration data are shown in Figure 3n-q. We find equal amount of human T cells in mice infused with LV-engineered and NILV-S/MAR-engineered CD19 CAR T cells.

Referee #3 (Comments on Novelty/Model System):

The authors have performed several experiments that were well performed and with adequate controls. Even if a lentiviral vector with s/MAR-containing sequences has been described the Authors showed that this system could have some potential for anticancer therapy in vivo.

Referee #3 (Remarks):

In the manuscript from Chuan Jin et al., the Authors describe the development of a vector platform based on integrase defective lentiviral vectors (IDLVs) containing a scaffold matrix attachment region (S/MAR) to enable episomal replication and long term transgene expression in proliferating cells. The possibility to achieve stable genetic engineering without integrating in the target cell genome would avoid the risks of insertional mutagenesis. The Authors convincingly show that the integration proficient LV and the IDLV-S/MAR designs allowed stable transgene expression in ex-vivo transduced cells (although with different efficiencies) while the IDLV without the S/MAR sequences failed to provide stable transgene expression. The Authors show that IDLV-S/MAR based vectors can be used to achieve stable knockdown of gene expression in cell lines. Finally, the Authors engineered T cells with a IDLV-S/MAR based vector encoding a 2nd generation CAR against the CD19 molecule and achieved stable CAR expression and biological effects in vitro and in vivo equivalent to those achieved with T-cells engineered with an integration proficient LV.

In summary the Manuscript shows that this platform is an interesting alternative platform for the safe genetic engineering of somatic cells in basic and translational applications.

Comments

The manuscript is interesting and, in general, the data is convincing. Moreover, the results obtained in vivo by the CD19 CAR-T cells are very encouraging.

The following issue however should be addressed:

Like for Adeno Associated Vectors (AAVs) and plasmids, it is possible that also the IDLV-S/MAR vectors integrate in the cell genome at low frequency using the cellular DNA repair machinery pathways. It will be interesting to estimate the levels (if any) of unwanted integration by this vector platform. The Authors used the specific PCR, lock-pad approach, and Linear Amplification Mediated (LAM) PCR to confirm the episomal status of the IDLV-S/MAR vector and to exclude genomic integration. Unfortunately none of these approaches fully address these issues.

a) Vector specific PCRs designed to amplify one or two LTR sequences present in the circularized episomal forms cannot be used to exclude integration of vectors that integrate by DNA repair based mechanisms because the circularized DNA forms can integrate and still retain the two or single LTR sequences.

We appreciate the reviewer's comments. It is correct that a specific PCR designed to amplify one or two LTR sequences present in the circularized episomal forms cannot be used to exclude integration of vectors. It only proves the existence of circularized DNA in IDLV-S/MAR transduced cells. We apologize if our text indicated otherwise. The text in the new version of the manuscript has been changed and does not.

LAM-PCR and Southern Blot (which we have added) are the two assays that can be used to detect integration and for LV-transduced cells we demonstrate integration with these methods.

We did not detect integration for NILV or NILV-S/MAR. This does not prove that integration does not occur but if it does it is below the detection level of these two assays.

Taking all experiments into consideration we can conclude that the vast majority of transgene expression from NILV-S/MAR transduced cells is maintained as circularized DNA with integration events below detection level (Figure 1e). We have made sure to re-phase the interpretation of results throughout the text.

b) The specificity of the lock-pad assay is questionable. While the cells treated with the IDLV without S/MAR sequences do not show any signal in their nuclei (as expected), the integration proficient LV shows staining levels similar to those found in the nuclei of IDLV-S/MAR cells after puromycin selection. These results suggest that the signal seen in cells treated with the integrase proficient LV arise from integrated forms since its episomal forms, lacking the S/MAR sequences like the IDLV control, should be lost when cells proliferate. Therefore it is not possible to formally conclude that the signals seen in the IDLV-S/MAR treated cells are specific for episomal forms.

We agree with the reviewer that the Pad-Lock assay cannot distinguish between integration events and episomal events. We apologize if our text could be interpreted that way, which was not our intention. As, pointed out by the reviewer, since the NILV control lacks signal the LV signals must come from integration events. It is therefore interesting to conclude that the amount of signals is equally high for the NILV-S/MAR vector.

We have made sure to re-phase the interpretation of results throughout the text.

c) The patterns of LAM PCR products obtained from the different conditions (LV, IDLV-S/MAR and IDLV) showed a smear of bands only when cells were treated with the integrase proficient LV. Unfortunately this technique is well suited to retrieve vector/cellular genomic junctions of vectors that integrate precisely like those generated by the active LV integrase. On the contrary, LAM PCR is inefficient in retrieving the vector/genome junctions of the integrated AAVs or plasmids that do not integrate in a precise manner and thus do not create specific vector ends like those generated by the LV integrase. Therefore it is not a surprise that the LAM PCR bands in the agarose gel electrophoresis appear only when the cells are treated with the integration proficient LV.

To measure the levels of unwanted integration (if detectable) the Authors could use a Southern blot strategy in which the DNA from cells transduced with the different vectors (LV, IDLV-S/MAR and IDLV) and kept in culture without selective pressure is digested with a restriction enzyme that cut only once in the vector genome and probed for vector sequences. Digested circular episomal vector forms will be linearized and display a specific size while the integrated vector forms will display a different size depending on the position of the restriction site on the host cell and vector genomes.

We thank the reviewer for this correct comment about the LAM-PCR and suggestion to use Southern Blot. We have now performed Southern Blot to prove that the NILV-SMAR vector maintains as circularized DNA forms with integration events below detection levels (Figure 1e). For the integrated LV vectors we did not obtain a specific band, most likely due to the fact that there are multiple integrations in one cell, possibly also indicated by the high MFI

values from flow cytometry experiments for LV-transduced cells in Figure 1b and supplementary Figure 1b.

Minor issue

The Authors fail to mention that the novel gene therapy vectors have an advanced design with self-inactivating LTRs and have an improved safety profile with respect those used previously. This of course does not means that non integrating vectors are not a significant safety improvement in the gene therapy field. However, given the wide use of integrating LV in gene therapy it would be fair to provide a more balanced literature background.

We thank the reviewer for this correct comment. We have added discussion about self-inactivating LTRs in the introduction of the manuscript.

Thank you for the submission of your revised manuscript to EMBO Molecular Medicine. We have now received the enclosed reports from the referees that were asked to re-assess it.

Two reviewers are now globally supportive while Reviewer 2 has some remaining concerns. Although I am not asking you to provide further experimentation at this stage, I would ask you to carefully address the Reviewer's points. Should you have further data available however, I would encourage you to include these in the revised manuscript. Also, please remove the previous red lettering from the manuscript and highlight the final amendments in the revised manuscript. Provided you fully address the reviewer's concerns, I will make an editorial decision on your manuscript.

Furthermore, please also take care of the following final amendments:

1) As per our Author Guidelines, the description of all reported data that includes statistical testing must state the name of the statistical test used to generate error bars and P values, the number (n) of independent experiments underlying each data point (not replicate measures of one sample), and the actual P value for each test (not merely 'significant' or ' $P < 0.05$ '). If necessary or preferred, you may add an additional appendix table to list all the P values, in which case, please make sure the manuscript is modified accordingly with the appropriate callouts!

2) We are now encouraging the publication of source data, particularly for electrophoretic gels and blots, with the aim of making primary data more accessible and transparent to the reader. Would you be willing to provide a PDF file per figure that contains the original, uncropped and unprocessed scans of all or at least the key gels used in the manuscript? The PDF files should be labeled with the appropriate figure/panel number, and should have molecular weight markers; further annotation may be useful but is not essential. The PDF files will be published online with the article as supplementary "Source Data" files. If you have any questions regarding this just contact me.

3) Every published paper now includes a 'Synopsis' to further enhance discoverability. Synopses are displayed on the journal webpage and are freely accessible to all readers. They include a short standfirst as well as 2-5 one sentence bullet points that summarise the paper. Please provide the synopsis including the short list of bullet points that summarise the key NEW findings. The bullet points should be designed to be complementary to the abstract - i.e. not repeat the same text. We encourage inclusion of key acronyms and quantitative information. Please use the passive voice. Please attach this information in a separate file or send them by email, we will incorporate it accordingly. You are also welcome to suggest a striking image or visual abstract to illustrate your article. If you do please provide a jpeg file 550 px-wide x 400-px high.

Please submit your revised manuscript within two weeks. I look forward to seeing a revised form of your manuscript as soon as possible.

***** Reviewer's comments *****

Referee #1 (Comments on Novelty/Model System):

Authors provide a strategy for safe and efficient transfer CAR expression into lymphocytes. Experiments are done properly and their results are clear and reliable.

Referee #1 (Remarks):

Authors have addressed all the required points for improvement of the manuscript. It is now suitable for publication in EMBO Mol Med.

Referee #2 (Comments on Novelty/Model System):

The authors have performed more efficacy experiments. However, the results are not wholly convincing. More convincing anti-tumour data in vivo would be reassuring.

Referee #2 (Remarks):

In this re-revised manuscript, the authors present new data to address comments from reviewers.

For consistency, this reviewer will return to the two comments made in both the 2014 and 2015 reviews.

1. Demonstration of non-integration. In this version of the manuscript, the authors present a new Southern blot, suggesting that there is a single species in two NILV-S/MAR transduced clones that is not detectable in two separate LV-transduced cells clones. The implication is that there are multiple insertion sites in the LV-transduced cells and thus single EcoR1 digestion will not liberate a single species to hybridize at 6.8KB. The authors have also updated the annotations of the LAM-PCR experiment (now Fig. S3). I still maintain that the Pad-lock assay does not demonstrate absence of integration, but overall, the authors have made good attempts to demonstrate that the NILV-S/MAR are non-integrating. Moreover, the results section states clearly that any integration events are below the limit of detection, which this reviewer endorses.

2. Anti-tumour effect. New experiments have been performed with 12 animals per group (Fig. 3i-m). Overall, the results are not striking or outstanding. There is still relentless tumour growth in all groups and I am puzzled that there is a significant difference in tumour volume on day 37 given the overlapping error bars. Presentation of 'survival' in subcutaneous experiments is always difficult given that the results actually represent 'time to reach defined measurements'. A statement that decisions about who made decisions about when animals were killed would be reassuring (assuming that it was animal house staff who made those decisions).

So, overall, the authors have made more efforts to demonstrate lack of integration. The in vivo data are far from impressive, however.

Referee #3 (Remarks):

The Authors answered satisfactorily to the Reviewer's comments

2nd Revision - authors' response

04 April 2016

Point-to-point response to reviewers

1. Demonstration of non-integration. In this version of the manuscript, the authors present a new Southern blot, suggesting that there is a single species in two NILV-S/MAR transduced clones that is not detectable in two separate LV-transduced cells clones. The implication is that there are multiple insertion sites in the LV-transduced cells and thus single EcoR1 digestion will not liberate a single species to hybridize at 6.8KB. The authors have also updated the annotations of the LAM-PCR experiment (now Fig. S3). I still maintain that the Pad-lock assay does not demonstrate absence of integration, but overall, the authors have made good attempts to demonstrate that the NILV-S/MAR are non-integrating. Moreover, the results section states clearly that any integration events are below the limit of detection, which this reviewer endorses.

We agree with reviewer's comment. We are fully aware that padlock assay does not exclude integration events and that it only tell whether the DNA persists. We had revised our text in the last version of the manuscript and we now highlight it again in red.

We appreciate the reviewer's remark that we now state that any integration events are below the limit of detection.

2. Anti-tumour effect. New experiments have been performed with 12 animals per group (Fig. 3i-m). Overall, the results are not striking or outstanding. There is still relentless tumour growth in all groups and I am puzzled that there is a significant difference in tumour volume on day 37 given the overlapping error bars. Presentation of 'survival' in subcutaneous experiments is always difficult given that the results actually represent 'time to reach defined measurements'. A statement that decisions about who made decisions about when animals were killed would be reassuring (assuming that it was animal house staff who made those decisions).

We agree with the reviewer that the *in vivo* results are not striking but can only conclude that the T cells engineered with NILV-S/MAR vector are equally effective as conventional lentivirus engineered T cells.

We do observe $p < 0.05$ when comparing tumor volume on day 37 using One-way ANOVA with Bonferroni correction. The error bar showed SD and it is hard to draw conclusion based on overlapping of SD error bar. This figure is now updated with error bar showing SEM. This is more illustrative and in accordance with the EMBO journal author's guidelines. The statistical methods are reported in M&M section as well as in the figure legends. All p values (or adjusted p values) for all comparisons are reported in a new Appendix Table S1.

Euthanasia of mice with s.c. tumors is defined in M&M part in page 22: "The mice were sacrificed either when the tumor volume exceeded 1 cm^3 or tumors became ulcerous". The tumor volume was calculated using the following formula $V = L * W^2 * \pi / 6$.

The researchers who perform the practical work decide when the animal should be sacrificed, by consulting animal house staff and veterinarians (especially in the case of ulcerous tumors). The end-point criteria are in accordance with the European Union ethics regulations and approved by the local animal ethical committee. A sentence stating that animal house staff and veterinarians were consulted has been added.

Corresponding Author Name: Magnus Essand

Manuscript Number: EMM-2015-05869-V2